# SHORT OPTIMIZATION PATHS LEAD TO GOOD GENERALIZATION

## ABSTRACT

Optimization and generalization are two essential aspects of machine learning. In this paper, we propose a framework to connect optimization with generalization by analyzing the generalization error based on the length of optimization trajectory under the gradient flow algorithm after convergence. Through our approach, we show that, with a proper initialization, gradient flow converges following a short path with an explicit length estimate. Such an estimate induces a length-based generalization bound, showing that short optimization paths after convergence indicate good generalization. Our framework can be applied to broad settings. For example, we use it to obtain generalization estimates on three distinct machine learning models: underdetermined $\ell_p$ linear regression, kernel regression, and overparameterized two-layer ReLU neural networks.

## 1 INTRODUCTION

From the perspective of statistical learning theory, the goal of machine learning is to find a predictive function that can give accurate predictions on new data. For supervised learning problems, empirical risk minimization (ERM) is a common practice to achieve this goal. The idea of ERM is to minimize a cost function on observed data by an optimization algorithm. Therefore, a fundamental question is *whether an optimization algorithm produces a solution with good generalization*.

Recent works have shed light on providing theoretical explanations to this question from different angles. One line of works considered the case when gradient methods converge to minimal norm solutions (Bartlett et al., 2020; Tsigler & Bartlett, 2020; Liang & Rakhlin, 2020; Liang et al., 2020) on kernel regression, and then analyzed the generalization of those minimal norm solutions. However, the phenomenon of norm minimization has been known to happen only for the quadratic loss with an appropriate initialization. Another line of works focused on overparameterized models, e.g., neural networks under the Neural Tangent Kernel (NTK) regime (Allen-Zhu et al., 2019; Arora et al., 2019; Cao & Gu, 2020; Ji & Telgarsky, 2020; Chen et al., 2021), proving that overparameterized neural networks trained by (stochastic) gradient descent ((S)GD) have good generalization performance on certain target functions (for example, polynomial functions).

Although existing works have made significant progress on the interplay of optimization and generalization, they focused on studying specific models, such as the NTK and models possessing minimal norm solutions. In this paper, instead, we study general loss function conditions that induce interesting connections between optimization and generalization.

We start with a simple observation, as shown in Figure 1, that under a generic random initialization, the generalization performance for both linear regression model and random feature regression model is closely related to the length of the optimization path[1] after convergence. In particular, short optimization path are associated with good generalization. Here, by length we mean the trajectory or path length of the parameter evolution during training. This is not the "length of time" used for training, as is usually analyzed in early-stopping type of algorithms. Thus, the generalization error concerns the weights of model trained to completion by empirical risk minimization. This empirical investigation motivates us to use the trajectory length to connect optimization and generalization. Intuitively, the length of the optimization path can be viewed as a kind of capacity control, and a

---

[1] For discrete iterations, we use the sum of the distance between every two consecutive iterations to represent the length of the optimization path (See details in Appendix A.1).

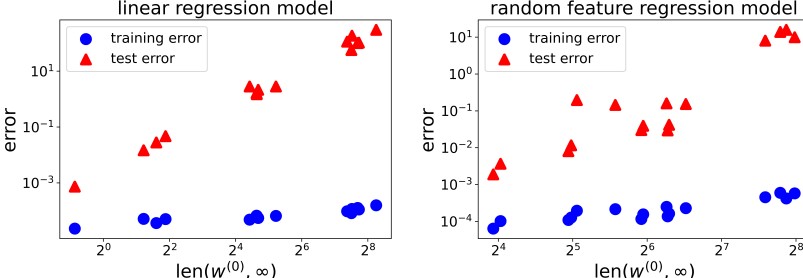

Figure 1: Illustrations of the relationship between optimization path lengths and generalization. **(Left)** Linear regression model. **(Right)** Random feature regression model with the feature extracted by a neural network. We train both models by gradient descent under Gaussian initialization $\mathcal{N}(\mu, \sigma^2)$ with varied $\mu$ and $\sigma$, and record the trajectory lengths $\mathsf{len}(w^{(0)}, \infty)$ from initialization until convergence. We observe that short optimization paths lead to good generalization. See experiment settings and more numerical results in Appendix A.1 & A.2.

short path signifies low "complexity". In other words, a general condition that guarantees a short optimization path can be used to induce good generalization performance. Thus, inspired by the theory in (Bolte et al., 2007) that Łojasiewicz gradient inequality (LGI) induces an explicit bound for the gradient flow path, we consider *Uniform-LGI* (Definition 1). This is a modified version of LGI and plays a critical role in obtaining a length estimate for the gradient flow trajectory. Once the length is estimated, we can use our length-based generalization bounds to show the generalization performance.

**Contributions.** We summarize our contributions as follows:

- We focus on the gradient flow algorithm and propose a framework for combining optimization and generalization. This framework is based on the length of the gradient flow trajectory, and its key component is the *Uniform-LGI* property.

- We then prove that under appropriate conditions, gradient flow returns a global minimum and gives an explicit length-estimate for the gradient flow trajectory (Theorem 1). If this length-estimate has an uniform bound for a certain initialization method, then we can give a length-based generalization bound (Theorem 2).

- We further show applications of Theorem 1 and Theorem 2 to obtain generalization bounds on underdetermined $\ell_p$ linear regression (Theorem 3), kernel regression (Theorem 4), and overparameterized two-layer ReLU neural networks (Theorem 5). These bounds match or expand the type of scenarios where we can rigorously establish the phenomenon of benign overfitting.

## 2    RELATED WORKS

**Optimization.** Theoretically analyzing the training process of most machine learning models is a challenging problem as the loss landscapes are highly non-convex. One approach to studying non-convex optimization problems is to use the Polyak-Łojasiewicz (PL) condition (Polyak, 1963), which characterizes the local geometry of loss landscapes and ensures the existence of global minima. It is shown in (Karimi et al., 2016) that GD admits linear convergence for a class of optimization objective functions under the PL condition. In this paper, we consider the Łojasiewicz gradient inequality (Lojasiewicz, 1965), an extended version of the PL condition that can be applied to more cases. In particular, we modify the original LGI to get length estimates of the optimization paths, which will be used to derive length-based generalization bounds.

**Generalization.** Traditional VC dimension-based generalization bounds depend on the number of parameters and will be vacuous for huge models such as overparameterized neural networks. To overcome this limitation, several non-vacuous generalization bounds are proposed. For example, the norm/margin-based generalization bounds (Neyshabur et al., 2015; Bartlett et al., 2017; Golowich

et al., 2018; Neyshabur et al., 2019), and the PAC-Bayes-based bounds (Dziugaite & Roy, 2017; Neyshabur et al., 2018; Zhou et al., 2019; Rivasplata et al., 2020). However, these bounds tend to focus less on optimization, e.g., norm-based generalization bounds may not discuss how small-norm solutions are obtained through practical training. In this paper, we connect the optimization and generalization by deriving generalization bounds based on the optimization trajectory length.

**Interface between optimization and generalization.** Implicit bias builds the bridge between optimization and generalization, which has been widely studied to explain the generalization ability of machine learning models. Recent works (Soudry et al., 2018a;b; Nacson et al., 2019a;b; Lyu & Li, 2020) showed that linear classifiers or deep neural networks trained by GD/SGD maximizes the margin of the separating hyperplanes and therefore generalizes well. Other works (Arora et al., 2019; Zou & Gu, 2019; Cao & Gu, 2020; Ji & Telgarsky, 2020; Chen et al., 2021) concentrated on overparameterized neural networks, showing that GD/SGD trajectories fall within the NTK regime, where the minimizer has good generalization due to the small "complexity" of the parameter space. In this work, we focus on specific conditions on loss functions under which we can connect optimization and generalization based on the lengths of optimization paths.

## 3 MAIN RESULTS

In this section, we aim to address optimization and generalization together by proving results based on the *Uniform-LGI* property. We begin with describing notations and the problem setting in Section 3.1. Then we give an explicit length estimate for the gradient flow trajectory in Section 3.2. Lastly, we show a length-based generalization bound in Section 3.3.

### 3.1 SETUP AND NOTATIONS

Consider a supervised learning problem on a hypothesis space $\mathcal{F} = \left\{ f(w, \cdot) : \mathbb{R}^d \to \mathbb{R} \mid w \in \mathcal{W} \right\}$, where $\mathcal{W}$ is a parameter set in Euclidean space. Given a loss function $\ell : \mathbb{R} \times \mathbb{R} \to \mathbb{R}$, and a training set $S = \{(x_i, y_i)\}_{i=1}^n \subseteq \mathbb{R}^d \times \mathbb{R}$ with $n$ independent and identically distributed (i.i.d.) samples from a joint distribution $\mathcal{D}$, the goal of ERM is to optimize the empirical loss function $\mathcal{L}_n(w)$ on $S$:

$$\arg \min_w \mathcal{L}_n(w) := \frac{1}{n} \sum_{i=1}^n \ell\left(f(w, x_i), y_i\right).$$

We assume that each input vector and label have bounded norms. Specifically, we assume that, for any $(x, y)$ following $\mathcal{D}$, $\|x\| = 1, |y| \leq 1$. This can be achieved by data normalization. To simplify the analysis, we optimize the empirical loss by gradient flow[2]:

$$\frac{dw^{(t)}}{dt} = -\nabla \mathcal{L}_n(w^{(t)}), \ t \in [0, +\infty), \tag{1}$$

where $w^{(t)}$ is the parameter value at time $t$, $w^{(0)}$ is the initialization parameter set, and $\nabla \mathcal{L}_n(w)$ is the gradient of $\mathcal{L}_n(w)$ with respect to $w$. Given the gradient flow, we may define the length of the gradient flow curve from $w^{(0)}$ to $w^{(t)}$ as $\mathsf{len}(w^{(0)}, t) := \int_0^t \left\| \frac{dw^{(s)}}{ds} \right\| ds$.

**Notations.** We use $\|\cdot\|$ to denote the $\ell_2$ norm of a vector or the spectral norm of a matrix, and use $\|\cdot\|_F$ to denote the Fronenius norm of a matrix. For a set $\mathcal{S} \in \mathbb{R}^n$, we use $\partial \mathcal{S}$ to denote its boundary. We use $d(\mathcal{S}_1, \mathcal{S}_2)$ to represent the Euclidean distance between two sets $\mathcal{S}_1, \mathcal{S}_2$, which is defined as $d(\mathcal{S}_1, \mathcal{S}_2) = \{\inf \|s_1 - s_2\| : s_1 \in \mathcal{S}_1, s_2 \in \mathcal{S}_2\}$. For two vectors, we use $\langle, \rangle$ to denote their inner product. Let $\lambda_{\min}(A)$ and $\lambda_{\max}(A)$ be the smallest and largest eigenvalues of a symmetric matrix $A$. For any positive integer $n$, we denote $[n] = \{1, 2, \ldots, n\}$. We use $\mathcal{O}(\cdot)$ to stand for Big-O notation.

The classical LGI gives a lower bound of the gradient of a differentiable function based on its value above its minimum. Many functions, e.g., real analytic functions and subanalytic functions, satisfy this property, at least locally (Bolte et al., 2007). Here, we require a global version of this inequality as a condition to control the gradient flow trajectory. Let us define this notion below.

**Definition 1** (Uniform-LGI). *A function $\mathcal{L}(w)$ satisfies Uniform-LGI on a set $\mathcal{S}$ with two constants $c > 0$ and $\theta \in [\frac{1}{2}, 1)$, if $\|\nabla \mathcal{L}(w)\| \geq c \left(\mathcal{L}(w) - \min_{v \in \mathcal{W}} \mathcal{L}(v)\right)^\theta$, $\forall w \in \mathcal{S}$.*

---

[2]Here we assume the existence of the gradient flow and the limit $\lim_{t \to \infty} w^{(t)}$.

In the special case when $\theta = 1/2$ and $c = \sqrt{2\mu}$, the Uniform-LGI corresponds to the $\mu$-PL condition (Karimi et al., 2016). Here we give examples of functions satisfying the Uniform-LGI but not the PL condition: $\mathcal{L}(w) = w^{2k}, k \in \mathbb{Z}^+$. When $k \geq 2$, $\mathcal{L}(w)$ satisfies the Uniform-LGI on $\mathbb{R}$ with $c = 2k$ and $\theta = 1 - 1/2k$.

## 3.2 GRADIENT FLOW TRAJECTORY LENGTH ESTIMATE

Our first result shows that under the uniform-LGI condition, gradient flow returns a global minimum with an explicit estimate of the gradient flow trajectory length as in the following theorem.

**Theorem 1.** *For any initialization $w^{(0)}$, if $\mathcal{L}_n(w)$ satisfies Uniform-LGI on a closed set $\mathcal{S}_n$ with constants $c_n, \theta_n$ such that $\mathcal{S}_n \supseteq B\left(w^{(0)}, r_n(w^{(0)})\right)$, where*

$$r_n(w^{(0)}) = \frac{\left(\mathcal{L}_n(w^{(0)}) - \min_w \mathcal{L}_n(w)\right)^{1-\theta_n}}{c_n(1-\theta_n)}, \tag{2}$$

*then $w^{(t)}$ converges to a global minimum $w^{(\infty)}$ with the trajectory length upper bounded by $\mathsf{len}(w^{(0)}, \infty) \leq r_n(w^{(0)})$. The convergence rate of $\mathcal{L}_n(w^{(t)})$ is given by:*

$$\theta_n = \frac{1}{2}: \qquad \mathcal{L}_n(w^{(t)}) - \min_w \mathcal{L}_n(w) \leq e^{-c_n^2 t}\left(\mathcal{L}_n(w^{(0)}) - \min_w \mathcal{L}_n(w)\right),$$

$$\frac{1}{2} < \theta_n < 1: \quad \mathcal{L}_n(w^{(t)}) - \min_w \mathcal{L}_n(w) \leq (1 + M_n t)^{-1/(2\theta_n - 1)}\left(\mathcal{L}_n(w^{(0)}) - \min_w \mathcal{L}_n(w)\right),$$

*where $M_n = c_n^2(2\theta_n - 1)\left(\mathcal{L}_n(w^{(0)}) - \min_w \mathcal{L}_n(w)\right)^{2\theta_n - 1}$.*

The proof of Theorem 1 is given in Appendix B.

This theorem yields that once the loss function satisfies the Uniform-LGI around the initialization, gradient flow returns a global minimum with an explicit trajectory length estimate. Moreover, for any fixed $n$, $\mathsf{len}(w^{(0)}, \infty)$ decreases as the initial loss value decreases. The calculation of $c_n, \theta_n$ for different $n$ should be analyzed case by case based on the loss landscape, as shown in Section 4.

## 3.3 HOW DOES THE OPTIMIZATION PATH LENGTH AFFECT GENERALIZATION?

The Uniform-LGI property allows us to have an upper bound for the gradient flow trajectory length. In this subsection, we use the length estimate we get in Theorem 1 to study generalization.

Throughout this subsection, we assume that there exists an almost everywhere differentiable function $\Psi : \mathbb{R}^{p+q} \to \mathbb{R}$ such that $f(w, \cdot)$ can be represented in the following form,

$$\forall x \in \mathbb{R}^d, \quad f(w, x) = \Psi\left(\alpha_1^\top x, \ldots, \alpha_p^\top x, \beta_1, \ldots, \beta_q\right) \tag{3}$$

with $\alpha_1, \ldots, \alpha_p \in \mathbb{R}^d$, $\beta_1, \ldots, \beta_q \in \mathbb{R}$, and $w = \mathsf{vec}\left(\{\alpha_1, \ldots, \alpha_p, \beta_1, \ldots, \beta_q\}\right) \in \mathbb{R}^{pd+q}$. Here vec is the vectorization operator that concatenates all elements into a column vector.

A wide class of functions can be represented in the form (3). Examples include linear functions $f(w, x) = w^\top x$ and two-layer neural networks $f(w, x) = a^\top \phi(Wx)$ with $a \in \mathbb{R}^m, W \in \mathbb{R}^{m \times d}$.

**Additional notations.** For the loss function $\ell$, we use $L_\ell(\mathcal{S})$ to denote its Lipschitz constant (the maximal gradient norm) on $\mathcal{S}$ with respect to its first argument. For $\Psi$ in (3), we define $L_\Psi(\mathcal{S}) := \left(L_\Psi^{(1)}(\mathcal{S}), \cdots, L_\Psi^{(p)}(\mathcal{S}), L_\Psi^{(p+1)}(\mathcal{S}), \cdots, L_\Psi^{(p+q)}(\mathcal{S})\right)^\top$, where $L_\Psi^{(i)}(\mathcal{S})$ is the Lipschitz constant of $\Psi$ on $\mathcal{S}$ with respect to the $i$-th variable. Let $w^{(0)} := \mathsf{vec}\left(\left\{\alpha_1^{(0)}, \ldots, \alpha_p^{(0)}, \beta_1^{(0)}, \ldots, \beta_q^{(0)}\right\}\right)$ and use $\mathcal{L}_\mathcal{D}(w)$ to denote the expected loss $\mathbb{E}_{(x,y) \sim \mathcal{D}}\left[\ell(f(w, x), y)\right]$. For $a = (a_1, \ldots, a_p)^\top$ and $b = (b_1, \ldots, b_q)^\top$, we define $\mathcal{S}_{a,b} = \{w : \forall i \in [p], j \in [q], \|\alpha_i\| \leq a_i, |\beta_j| \leq b_j\}$, and $M_{a,b} = \sup_{w \in \mathcal{S}_{a,b}, \|x\| \leq 1, |y| \leq 1} \ell(f(w, x), y)$.

In the following theorem, we give a path-based generalization bound if one has a length estimate for optimization trajectory length $\mathsf{len}(w^{(0)}, \infty)$. Based on the Uniform-LGI property, Theorem 1 serves as a sufficient condition to ensure that the trajectory length can be estimated. In that sense, we connect optimization with generalization. The rigorous statement is as follows:

**Theorem 2.** *Consider an initialization method so that $w^{(0)}$ is independent of the training samples and suppose that for any $\delta \in (0, 1)$, there exist $M_\delta, R_{n,\delta} > 0$ such that $\|w^{(0)}\| \leq M_\delta$, len$(w^{(0)}, \infty) \leq R_{n,\delta}$ with probability at least $1 - \delta$ over the initialization and the training samples. Then, we have with probability at least $1 - \delta$ over initialization and the training samples, the generalization error of the global minimum $w^{(\infty)}$ is bounded as:*

$$\mathcal{L}_\mathcal{D}(w^{(\infty)}) \leq \min_w \mathcal{L}_n(w) + \sup_{\|a\|^2 + \|b\|^2 \leq R^2} \frac{2RL_\ell(\mathcal{S}_{a,b}) \|L_\Psi(\mathcal{S}_{a,b})\|}{\sqrt{n}} + 3M_{a,b}\sqrt{\frac{3(p+q) + \log(2/\delta)}{2n}}, \tag{4}$$

*where $R = \sqrt{2}(M_\delta + R_{n,\delta})$.*

The proof of Theorem 2 is given in Appendix C. There are five key terms in (4). To make the bound easier to understand, we discuss them in sequence:

- $M_\delta$: This is a high probability upper bound for the $\ell_2$ norm of the initialized vector $\|w^{(0)}\|$. In practice, for commonly used initialization methods in deep learning, such as Xavier initialization (Glorot & Bengio, 2010) and Kaiming initialization (He et al., 2015), the norm $\|w^{(0)}\|$ is uniformly bounded with high probability.

- $R_{n,\delta}$: This is an high probability upper bound for the optimization path length over the initialization and training samples. Theorem 2 assumes that such a bound exists, and show that it immediately implies a generalization estimate based on this bound. Theorem 1 gives a sufficient condition to obtain such a path estimate, in which case it depends on the Uniform-LGI constants $c_n, \theta_n$ and data distribution $\mathcal{D}$. The asymptotic analysis of $R_{n,\delta}$ is problem-dependent, and we provide several examples in Section 4.

- $L_\ell(\mathcal{S}_{a,b})$ and $M_{a,b}$: They depend on the loss function $\ell$. For example, for any loss function $\ell : \mathbb{R} \times \mathbb{R} \to [0, 1]$ that is 1-Lipschitz in the first argument, $L_\ell(\mathcal{S}_{a,b}) = M_{a,b} = 1$.

- $\|L_\Psi(\mathcal{S}_{a,b})\|$: This term relies on the structure of $f$. For example, when $f$ is linear, $\Psi(x, y) = x + y$, then $\|L_\Psi(\mathcal{S}_{a,b})\| = \sqrt{2}$.

**Remark 1.** *Theorem 2 shows that if the optimization trajectory length is small, then this implies a good generalization bound. First, note that our generalization bound concerns the weights of the final model. Nevertheless, by the convergence analysis in Theorem 1, our framework can be extended to derive generalization estimates that evolves according the length of time of training. See Appendix C.1 for additional results. Second, we emphasize again that the generalization bound in Theorem 2 only relies on a path length estimate, and theorem 1 gives a sufficient condition to ensure that such an estimate exists. Consequently, as long as one can obtain path length estimates (say from stochastic analysis of SGD), one can still apply Theorem 2 to obtain generalization estimates. Path length estimates for other types of training algorithms is an interesting future direction.*

## 4 APPLICATIONS

In this section, we apply the framework to three models. To obtain clean expressions of the generalization bound in terms of the sample size $n$, it is helpful to consider a range of $n$ relating to the dimension $d$. In particular, we consider underdetermined systems where the ratio $n/d$ remains finite unless stated otherwise:

$$\exists \, \gamma_0, \gamma_1 \in (0, \infty), \; s.t. \, \forall d, \; \gamma_0 d \leq n = n(d) \leq \gamma_1 d.$$

Note that this setting is non-asymptotic since we do not require $d$ to be infinite. For each application, to simplify the analysis, we evaluate the test error on a loss function $\tilde{\ell} : \mathbb{R} \times \mathbb{R} \to [0, 1]$ is 1-Lipschitz (on the first argument) with $\tilde{\ell}(y, y) = 0$. The following steps are taken in our framework:

**Step 1.** Establish the Uniform-LGI property and find the Uniform-LGI constants $c_n$ and $\theta_n$.

**Step 2.** Apply Theorem 1 to get optimization results to estimate path length.

**Step 3.** Apply Theorem 2 to get generalization results from the path length estimates.

## 4.1 UNDERDETERMINED $\ell_p$ LINEAR REGRESSION

We begin with an underdetermined linear regression model $f(w, x) = w^\top x$ with an $\ell_p$ loss function ($p \geq 2$ is an even positive integer):

$$\arg\min_{w \in \mathbb{R}^d} \mathcal{L}_n(w) := \frac{1}{pn} \sum_{i=1}^{n} \left(w^\top x_i - y_i\right)^p, \tag{5}$$

where the input data matrix $\mathcal{X} = (x_1, \ldots, x_n)^\top \in \mathbb{R}^{n \times d}$ has full row rank. Then the above regression model has at least one global minimum with zero loss.

**Target function.** Suppose the training data is generated from an underlying function $g : \mathbb{R}^d \to \mathbb{R}$ with $y_i = g(x_i)$, $\forall i \in [n]$. Let $\mathcal{Y} = (y_1, \cdots, y_n)^\top$, and assume that there exits $c^* > 0$ such that

$$\|\mathcal{Y}\| \leq c^* \sqrt{\lambda_{\max}(\mathcal{X}\mathcal{X}^\top)}. \tag{6}$$

The inequality (6) actually indicates that $g$ is Lipschitz with a dimension independent Lipschitz constant. Functions satisfying (6) include linear/non-linear functions. For example, $g(x) = \phi(x^\top w^*)$, where $w^* \in \mathbb{R}^d$ with $\|w^*\|_2 \leq c^*$ for some constant $c^*$, and $\phi(\cdot)$ is Lipschitz with $\phi(0) = 0$.

**Assumption 1.** *The entries of $\mathcal{X}$ are i.i.d. subgaussian random variables with zero mean, unit variance, and subgaussian moments[3] bounded by $1$.*

This assumption allows us to study the spectral properties of the sample matrix by random matrix theory. Especially, when $n(d)/d$ converges to some constant $\gamma \in (0, 1)$, the Marchenko–Pastur law (Marčenko & Pastur, 1967) shows that $\lambda_{\min}(\mathcal{X}\mathcal{X}^\top)/n$ converges to $(1 - \sqrt{\gamma})^2$ almost surely. The non-asymptotic results are provided in (Rudelson & Vershynin, 2010).

Performing our three-step analysis, we get the following results:

**Theorem 3.** *Consider the undertermined $\ell_p$ linear regression model (5). Suppose that there exists a universal constant $c_0 \geq 1$ such that $\forall d$, $\left\|w^{(0)}\right\|_2 \leq c_0$.*

**Step 1.** *$\mathcal{L}_n(w)$ satisfies the Uniform-LGI globally on $\mathbb{R}^d$ with*

$$c_n = p^{1-1/p} \sqrt{\frac{\lambda_{\min}(\mathcal{X}\mathcal{X}^\top)}{n}}, \quad \theta_n = 1 - 1/p.$$

**Step 2.** *$\mathcal{L}_n(w^{(t)})$ converges to zero linearly for $p = 2$ and sublinearly for $p \geq 4$, i.e.,*

$$p = 2, \quad \mathcal{L}_n(w^{(t)}) \leq \exp\left(-2\lambda_{\min}(\mathcal{X}\mathcal{X}^\top)t/n\right) \mathcal{L}_n(w^{(0)});$$
$$p \geq 4, \quad \mathcal{L}_n(w^{(t)}) \leq (1 + Mt)^{-\frac{p}{p-2}} \mathcal{L}_n(w^{(0)}),$$

*where $M = p^{1-\frac{2}{p}}(p-2)\frac{\lambda_{\min}(\mathcal{X}\mathcal{X}^\top)}{n} \mathcal{L}_n(w^{(0)})^{1-\frac{2}{p}}$.*

**Step 3.** *Under Assumption 1, for any target function that satisfies (6), we have with probability at least $1 - \delta - \tau^{d-n+1} - \tau^d$ over the samples,*

$$\mathbb{E}_{(x,y)\sim\mathcal{D}}\left[\tilde{\ell}\left(f(w^{(\infty)}, x), y\right)\right] \leq \mathcal{O}\left(n^{-1/p}\right) + \mathcal{O}\left(\sqrt{\frac{\log(1/\delta)}{n}}\right),$$

*where $\tau \in (0, 1)$ depends only on the subgaussian moment of the entries.*

The proof of Theorem 3 is given in Appendix D.1. This theorem shows that compared to the PL condition that corresponds to $p = 2$, the uniform-LGI is more general and can be applied to more cases.

**Comparison.** This result is related to (Bartlett et al., 2020) that studied the phenomenon of benign overfitting in high-dimensional $\ell_2$ linear regression. Our result coincides with theirs as both results uncover some scenarios for benign overfitting in linear regression. In particular, (Bartlett et al.,

---

[3]The subgaussian moment of $X$ is defined as $\inf\left\{\mathcal{M} \geq 0 \mid \mathbb{E}e^{tX} \leq e^{\mathcal{M}^2 t^2/2}, \forall t \in \mathbb{R}\right\}$.

2020) focused on the minimum $\ell_2$ norm estimator. They showed that, if the eigenvalue sequence of the covariance operator $\Sigma := \mathbb{E}[xx^\top]$ have suitable decay rates, then the generalization error will decrease to zero as $n$ increases. In contrast, our result differs from theirs in the problem settings. Specifically, we do not assume the minimum norm property and consider the optimization process that is neglected in (Bartlett et al., 2020). Also, we evaluate the generalization error of the convergence point on a globally Lipschitz loss function. In our settings, entries of each input $x$ are i.i.d. random variables, meaning that $\Sigma = \mathbb{I}_{d \times d}$ without decaying eigenvalues. The requirement of i.i.d. entries for each input is a limitation due to the lack of non-asymptotic results of the spectral properties for random matrices with non-i.i.d. entries. However, this limitation can be overcome if one considers the asymptotic results. Moreover, our result works for $\ell_p$ loss functions with any even positive integer $p$. This expands the understanding of benign overfitting in the high-dimensional setting.

## 4.2 KERNEL REGRESSION

Consider a positive definite kernel $k : \mathcal{X} \times \mathcal{X} \to \mathbb{R}$ with its corresponding feature map $\varphi : \mathbb{R}^d \to \mathcal{F}$ satisfying $\langle \varphi(x), \varphi(y) \rangle_{\mathcal{F}} = k(x, y)$. We assume that $|k(x, x)| \leq 1, \forall x \in \mathcal{X}$. Let $\mathcal{H}$ be the reproducing kernel Hilbert space (RKHS) with respect to $k$. If $\mathcal{F} = \mathbb{R}^s$, then the kernel regression model with $\ell_p$ loss is to solve the following problem

$$\underset{w \in \mathbb{R}^s}{\arg\min} \, \mathcal{L}_n(w) := \frac{1}{2n} \sum_{i=1}^n \left( w^\top \varphi(x_i) - y_i \right)^p, \tag{7}$$

where $p \geq 2$ is an even integer. Similar to the $\ell_p$ linear regression case, we consider the following target function class:

**Target function.** Suppose the training data is generated from an underlying function $g : \mathbb{R}^d \to \mathbb{R}$ with $y_i = g(x_i)$, $\forall i \in [n]$. We further assume that there exits a constant $c^* > 0$ such that

$$\|\mathcal{Y}\| \leq c^* \cdot \sqrt{\lambda_{\max}(k(\mathcal{X}, \mathcal{X}))}, \tag{8}$$

where $k(\mathcal{X}, \mathcal{X})$ is the $n \times n$ kernel matrix on $\mathcal{X}$ with $k(\mathcal{X}, \mathcal{X})_{ij} = k(x_i, x_j)$.

Here, we list an example of class of functions that satisfies (8): $g(x) = \phi(\varphi(x)^\top w^*)$ where $w^* \in \mathbb{R}^s$ with $(\forall s) \|w^*\|_2 \leq c^*$ for some constant $c^*$, and $\phi(\cdot)$ is Lipschitz with $\phi(0) = 0$.

To get the generalization results of kernel regression, we will discuss two types of kernels separately: radial basis function (RBF) kernel and inner product kernel.

**RBF kernel.** We study the RBF kernel of the form $k(x, y) = \varrho(\|y - x\|)$ for a certain RBF $\varrho$. For the input data, we define the *separation distance* of $\mathcal{X}$ as $\mathsf{SD} := \frac{1}{2} \min_{i \neq j} \|x_i - x_j\|, \forall i, j \in [n]$.

**Inner product kernel.** For the inner product kernel, we consider $k(x, y) = \varrho\left(\frac{x^\top y}{d}\right)$.

Following (El Karoui et al., 2010), we make the following assumption on the function $\varrho$:

**Assumption 2.** $\varrho$ is $C^3$ in a neighborhood of $0$ with $\varrho(0) = 0$, $\varrho(1) > \varrho'(0) \geq 0$, $\varrho''(0) \geq 0$.

Now we are ready to apply our three-step analysis to get optimization and generalization results. For the RBF kernel, the generalization result depends on the separation distance of the samples. For the inner product kernel, we study the high-dimensional random kernel matrix.

**Theorem 4.** *Consider the kernel regression model (7). Suppose that there exists a universal constant $c_0 \geq 1$ such that $\forall s$, $\|w^{(0)}\|_2 \leq c_0$.*

**Step 1.** $\mathcal{L}_n(w)$ *satisfies the Uniform-LGI globally on $\mathbb{R}^s$ with*

$$c_n = p^{1-1/p} \sqrt{\frac{\lambda_{\min}(k(\mathcal{X}, \mathcal{X}))}{n}}, \quad \theta_n = 1 - 1/p,$$

*where $c_n$ is controlled by the kernel and input samples.*

**Step 2.** $\mathcal{L}_n(w^{(t)})$ *converges to zero linearly for $p = 2$ and sublinearly for $p \geq 4$, i.e.,*

$$p = 2, \quad \mathcal{L}_n(w^{(t)}) \leq \exp\left(-2\lambda_{\min}(k(\mathcal{X}, \mathcal{X}))t/n\right) \mathcal{L}_n(w^{(0)});$$

$$p \geq 4, \quad \mathcal{L}_n(w^{(t)}) \leq (1 + Mt)^{-\frac{p}{p-2}} \mathcal{L}_n(w^{(0)}),$$

where $M = p^{1-\frac{2}{p}} (p-2) \frac{\lambda_{\min}(k(\mathcal{X}, \mathcal{X}))}{n} \mathcal{L}_n(w^{(0)})^{1-\frac{2}{p}}$.

**Step 3.** *For any target function that satisfies (8) we have:*

- *For the RBF kernel[4], suppose that $\varrho : \mathbb{R}_{\geq 0} \to \mathbb{R}_{\geq 0}$ is a decreasing function and $\varrho (\|x\|) \in L^1(\mathbb{R}^d)$. If there exists two positive constants $q_{\min}$ and $q_{\max}$ such that $\mathsf{SD} \in [q_{\min}, q_{\max}]$ for all $n$, then with probability at least $1 - \delta$ over the samples,*

$$\mathbb{E}_{(x,y) \sim \mathcal{D}} \left[ \tilde{\ell} \left( f(w^{(\infty)}, x), y \right) \right] \leq \mathcal{O} \left( n^{-1/p} \right) + \mathcal{O} \left( \sqrt{\frac{\log(1/\delta)}{n}} \right).$$

- *For the inner product kernel, under Assumption 1 and Assumption 2, if $d$ is large enough and $\delta > 0$ is small enough such that $d^{-1/2} \left( \sqrt{3} \delta^{-1/2} + \log^{0.51} d \right) \leq 0.5(\varrho(1) - \varrho'(0))$, then with probability at least $1 - \delta - d^{-2}$ over the samples, we have*

$$\mathbb{E}_{(x,y) \sim \mathcal{D}} \left[ \tilde{\ell} \left( f(w^{(\infty)}, x), y \right) \right] \leq \mathcal{O} \left( n^{-1/p} \right) + \mathcal{O} \left( \sqrt{\frac{\log(1/\delta)}{n}} \right).$$

The proof of Theorem 4 is given in Appendix D.2.

**Example 1.** *RBF kernels satisfying the conditions and assumptions in Theorem 4 include (1) Gaussian: $\varrho(r) = e^{-\rho r^2}, \rho > 0$; (2) (inverse) Multiquadrics: $\varrho(r) = (\rho + r^2)^{\beta/2}, \rho > 0, \beta \in \mathbb{R} \backslash 2\mathbb{N}, \beta < -d$.*

*Inner product kernels satisfying the conditions and assumptions in Theorem 4 include (1) Polynomial kernel: $\varrho(r) = r^\beta, \beta \in \mathbb{Z}^+, \beta \geq 2$; (2) NTK corresponding to Two-layer ReLU neural networks on $\mathbb{S}^{d-1}(\sqrt{d})$: $\varrho(r) = \frac{r(\pi - \arccos(r))}{2\pi}$.*

**Comparison.** The $p = 2$ result of the inner product kernel is related to (Liang & Rakhlin, 2020) who derived generalization bounds for the minimum RKHS norm estimator. They showed that when the data covariance matrix and the kernel matrix enjoy certain decay of the eigenvalues, the generalization bound vanishes as $n$ goes to infinity. For example, for exponential kernel and the covariance matrix $\Sigma := \mathbb{E}[xx^\top]$ with the $j$-th eigenvalue $\lambda_j(\Sigma) = j^{-\alpha}$, the $\ell_2$ generalization bound becomes $\mathcal{O}(n^{-\frac{\alpha}{2\alpha+1}})$ when $\alpha \in (0, 1)$ and $n > d$. In comparison, we get an optimal generalization bound $\mathcal{O}\left(n^{-1/2}\right)$ for identity covariance matrix on a globally Lipschitz loss function. Again, we emphasize that the i.i.d. assumption can be relaxed if more new results of random matrix theory are available. Further, we extend the works in (Liang & Rakhlin, 2020) by proving a new result of the RBF kernel. Note that the result of the RBF kernel is not under the high-dimensional setting; thus it is not a direct adaptation of (Liang & Rakhlin, 2020), and the proof itself is of independent interest.

### 4.3 OVERPARAMETERIZED TWO-LAYER NEURAL NETWORKS

The first two applications are on traditional machine learning models. Indeed, our framework can be applied not only to linear/kernel regression but also to neural network models. In this subsection, we use our framework to study shallow neural networks in an overparameterization regime.

First, define a two-layer ReLU neural network under the *NTK parameterization* (Lee et al., 2019):

$$f(w, x) = \frac{1}{\sqrt{m}} w_2^\top \phi(W_1 x),$$

where $\phi(x) = \max\{0, x\}$, $x \in \mathbb{R}^d$ is the input, $W_1 \in \mathbb{R}^{m \times d}$, $w_2 \in \mathbb{R}^m$ are parameters, $w = \text{vec}\left(\{W_1, w_2\}\right) \in \mathbb{R}^{m(d+1)}$, and $m$ is the width (number of hidden units).

We consider minimizing the quadratic loss by gradient flow:

$$\arg \min_w \mathcal{L}_n(w) := \frac{1}{2n} \sum_{i=1}^n (f(w, x_i) - y_i)^2. \tag{9}$$

---

[4]Here $d$ is fixed and $n$ is varied.

**Random initialization.** $W_1^{(0)}$ is drawn from Gaussian $\mathcal{N}(0, \frac{1}{d}\mathbb{I}_{m \times d})$ and $w_{2,i}^{(0)}$ are drawn i.i.d. from uniform distribution $U\{-1, 1\}, \forall i \in [m]$.

Following the setting in (Du et al., 2019), we only train the hidden layer $W_1$ and leave the output layer $w_2$ as random initialization to simplify analyses.

**NTK matrix.** The NTK matrix $\Theta(t)$ is defined as: $\Theta_{ij}(t) = \langle \nabla_{W_1} f(w^{(t)}, x_i), \nabla_{W_1} f(w^{(t)}, x_j) \rangle$, and denote $\widehat{\Theta}$ by the limiting matrix[5]: $\widehat{\Theta}_{ij} = x_i^\top x_j \mathbb{E}_{w \sim \mathcal{N}(0, \frac{1}{d}\mathbb{I}_d)} [\phi'(w^\top x_i)\phi'(w^\top x_j)], \forall i, j \in [n]$.

Similarly, we apply our three-step analysis to derive the following results.

**Theorem 5.** *Consider the two-layer ReLU neural network model (9). For any* $\delta \in (0, 1)$, *if* $m \geq \text{poly}\left(n, \lambda_{\min}^{-1}(\widehat{\Theta}), \delta^{-1}\right)$, *then*

***Step 1.*** *With probability at least* $1 - \delta$ *over training samples and random initialization,* $\mathcal{L}_n(w^{(t)})$ *satisfies the Uniform-LGI on* $\{w^{(t)} : t \geq 0\}$ *with*

$$c_n = \sqrt{\lambda_{\min}(\widehat{\Theta})/n}, \quad \theta_n = 1/2.$$

***Step 2.*** $\mathcal{L}_n(w^{(t)})$ *converges to zero with a linear convergence rate:*

$$\mathcal{L}_n(w^{(t)}) \leq \exp\left(-\lambda_{\min}(\widehat{\Theta})t/n\right) \mathcal{L}_n(w^{(0)}).$$

***Step 3.*** *Under Assumption 1, for any target function that satisfies (6), if* $\gamma_1 \in (0, 1)$, *then with probability at least* $1 - \delta - \tau^{d-n+1} - \tau^d$ *over the samples and random initialization,*

$$\mathbb{E}_{(x,y) \sim \mathcal{D}}\left[\tilde{\ell}\left(f(w^{(\infty)}, x), y\right)\right] \leq \mathcal{O}\left(\sqrt{\frac{\log(n/\delta)}{n}}\right),$$

*where* $\tau \in (0, 1)$ *depends only on the subgaussian moment of the entries.*

The proof of Theorem 5 is deferred to Appendix D.3.

**Comparison.** Our result is related to (Arora et al., 2019), which gave an NTK-based generalization bound for overparameterized two-layer ReLU neural networks. This result matches with theirs in the sense that we discover some underlying functions that are provably learnable. Examples of learnable target functions in (Arora et al., 2019) include polynomials $y = (\beta^\top x)^p$, non-linear activations $y = \cos(\beta^\top x) - 1$, $y = \tilde{\phi}(\beta^\top x)$ with $\tilde{\phi}(z) = z \cdot \arctan(z/2)$, $\|\beta\| \leq 1$. Our result, furthermore, expands the target function class that is provably learnable since we only require $\tilde{\phi}$ to be Lipschitz. In addition, they set the standard deviation of the initialization to be at most $\mathcal{O}(1/n)$, whereas we use a different initialization with order $\mathcal{O}(1/\sqrt{d})$ that is more often applied in practice. Our result is also related to (Liu et al., 2020), which proved that overparameterized deep neural networks satisfy the PL condition. Further, we extend this work by analyzing the generalization based on the length of optimization trajectories.

## 5 CONCLUSION

In this work, we address when and why does an optimization algorithm finds a minimum with good generalization performance. For this problem, we propose a framework to bridge optimization and generalization based on the trajectory length. The pivotal component is the Uniform-LGI property: a condition on the loss function, by which we show that gradient flow returns a global minimum with an explicit length estimate. Further, we derive a length-based generalization bound given such a length estimate. Finally, we apply the framework to three machine learning models with certain target functions. By estimating the trajectory lengths, we get non-vacuous generalization bounds that do not suffer from the curse of dimensionality. This framework is not a direct variant of the NTK method, and the results show that our framework is favorable for inducing connections between optimization and generalization.

---

[5]Here $\lambda_{\min}(\widehat{\Theta})$ changes with $n$.

## 6 REPRODUCIBILITY STATEMENT

For the experiments, details about the models and the synthetic data are described in Appendix A.1 and Appendix A.2. For the theoretical results, we have included clear explanations of all assumptions. The complete proofs are provided in Appendix B, C & D.

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

# A    EXPERIMENTS

## A.1    EXPERIMENTS SETTINGS

In this subsection, we describe the details of our experiments in the introduction part (Figure 1).

**Linear Regression model.** The model that we use is underdetermined $\ell_2$ linear regression without regularization. The data set consists of 130 data points $\{(x_i, y_i)\}_{i=1}^{130} \subseteq \mathbb{R}^{200} \times \mathbb{R}$. The inputs $x_i (\forall i \in [130])$ are uniformly drawn from the 200-dimensional unit sphere. All labels $y_i (\forall i \in [130])$ are generated by a linear target function $y_i = \beta^\top x_i$ with some $\beta \in \mathbb{R}^{200}$ that satisfies $\|\beta\| = 1$. We train this model by gradient descent $w^{(k+1)} = w^{(k)} - \eta \nabla \mathcal{L}_n(w^{(k)})$ with step size $\eta = 0.05$ on the mean square loss (MSE). Entries of $w^{(0)}$ are initialized i.i.d. from Gaussian distribution $\mathcal{N}(\mu, \sigma^2)$ with $\mu = 2^{-5}, 2^{-2}, 2^1, 2^4$ and $\sigma = 2^{-5}, 2^{-2}, 2^1, 2^4$ respectively. We stop the training once the difference of loss between two consecutive steps $|\mathcal{L}_n(w^{(K+1)}) - \mathcal{L}_n(w^{(K)})| < 10^{-8}$. Then $\text{len}(w^{(0)}, \infty)$ is approximated by $\sum_{k=1}^{K} \|w^{(k)} - w^{(k-1)}\|$. For each pair of $\mu$ and $\sigma$, we record the length $\text{len}(w^{(0)}, \infty)$, the training error after convergence and the test error on the convergence point.

**Random feature model.** In this experiment, the model that we consider is a two-layer ReLU neural network $w_2^\top \phi(W_1 x)$. Here $\phi$ is the ReLU function, $W_1 \in \mathbb{R}^{200 \times 200}, w_2 \in \mathbb{R}^{200}$, We only train the last layer $w_2$, and thus this can be viewed as a random feature model with features extracted by $\phi(W_1 x)$. The data set consists of 130 data points $\{(x_i, y_i)\}_{i=1}^{130} \subseteq \mathbb{R}^{200} \times \mathbb{R}$. The inputs $x_i (\forall i \in [130])$ are uniformly drawn from the 200-dimensional unit sphere. All labels $y_i (\forall i \in [130])$ are generated by a teacher network $\widehat{w}_2^\top \phi(\widehat{W}_1 x_i)$ with the same architecture for some $\widehat{w}_2, \widehat{W}_1$. We train only the top layer by gradient descent with momentum 0.9 and step seize 0.05 under the mean square loss (MSE). Entries of $w_2^{(0)}$ are initialized i.i.d. from $\mathcal{N}(\mu, \sigma^2)$ with $\mu = -5, -2, 1, 4$ and $\sigma = 2^{-5}, 2^{-2}, 2^1, 2^4$ respectively. We stop the training once the difference of loss between two consecutive steps is less than $10^{-8}$. For each pair of $\mu$ and $\sigma$, we use the same method as on linear regression to approximate the trajectory length. Then we record the training error after convergence and the test error on the convergence point.

**Varied learning rate.** In this experiment, we provide additional results by only changing the learning rate. Figure 2, 3 and 4 correspond to step size $\eta = 0.01, 0.1, 0.5$ respectively. We observe that the optimization path lengths are nearly the same for both small learning rate and moderate learning rate, and the numerical results are associate with our theory.

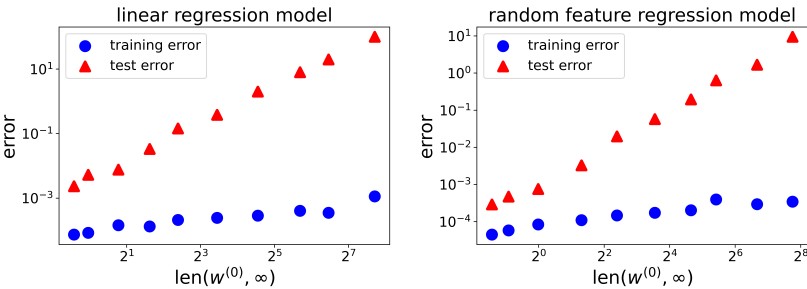

Figure 2: learning rate = 0.01

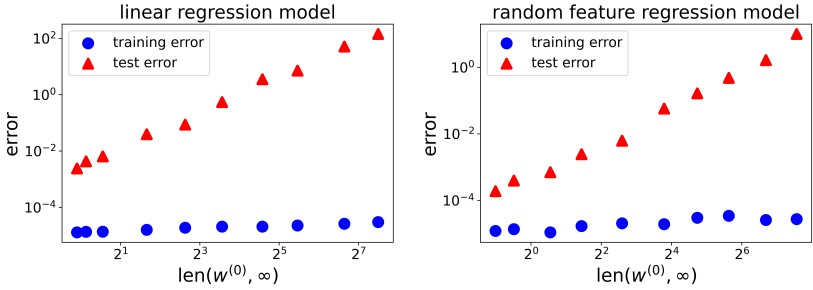

Figure 3: learning rate = 0.1

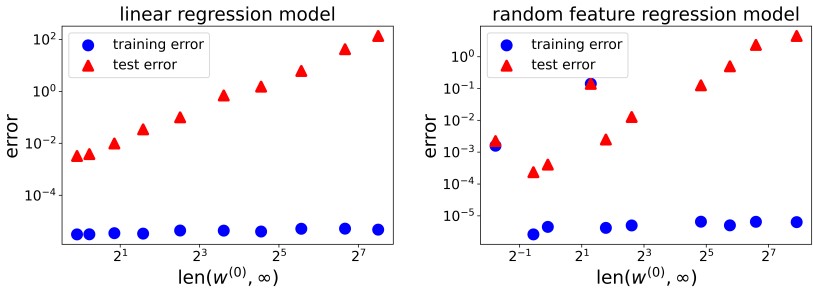

Figure 4: learning rate = 0.5

## A.2 MNIST CLASSIFICATION

In this subsection, we calculate the trajectory length on the MNIST classification problem to show the relation between optimization and generalization under different initializations.

**Experiment settings.** In this experiment, we train a two-layer ReLU neural network with 100 hidden units. We use stochastic gradient descent with mini-batch 64, momentum 0.9 and initial learning rate 0.1 to train the model. Parameters of the hidden layer are initialized by standard random initialization method in Pytorch, and the parameters of the top layer are initialized i.i.d. from Gaussian distribution $\mathcal{N}(\mu, \sigma^2)$ with $\mu = 2^{-5}, 2^0, 2^5$ and $\sigma = 2^{-5}, 2^{-2}, 2^1, 2^4$ respectively. During training the model, we reduce the learning rate by a factor 0.1 once learning stagnates. We stop the training once the cross-entropy loss decreases to 0.01 or the number of epochs reaches 1000. For each pair of $\mu$ and $\sigma$, we calculate the trajectory length by $\sum_{k=1}^{K} \left\| w^{(k)} - w^{(k-1)} \right\|$. We plot the relation between optimization and generalization in terms of the trajectory length in Figure 5.

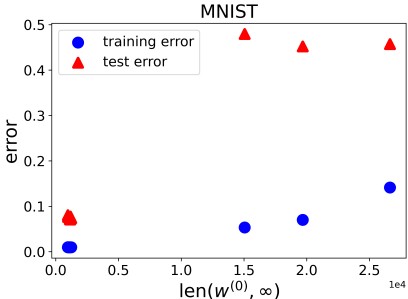

Figure 5: MNIST classification

From this figure, we can see that for different random initializations, short optimization paths is associated with good generalization gap. This numerical result exhibits the generality of our theory and suggests that the path length plays an important role to connect optimization and generalization.

## B   PROOF OF THEOREM 1

In this section we will prove Theorem 1, our proof is based on the next lemma, showing that the gradient flow trajectory is always inside $\mathcal{S}_n$.

**Lemma B.1.** *For any initialization $w^{(0)}$, if $\mathcal{L}_n(w)$ satisfies Uniform-LGI on a closed set $\mathcal{S}_n \supseteq B\left(w^{(0)}, r_n(w^{(0)})\right)$ with constants $c_n, \theta_n$, where $r_n(w^{(0)}) = \frac{\left(\mathcal{L}_n(w^{(0)}) - \min_w \mathcal{L}_n(w)\right)^{1-\theta_n}}{c_n(1-\theta_n)}$, then*

$$w^{(t)} \in \mathcal{S}_n, \ \forall t \in [0, \infty).$$

*Proof.* Let

$$T = \inf\{t \geq 0, \ w^{(t)} \notin \mathcal{S}_n\},$$

then it is sufficient to prove that $T = \infty$. Otherwise, if $T < \infty$, then by the continuity of the curve $\{w^{(t)}\}_{t \geq 0}$, we know that $w^{(T)}$ is in the boundary of $\mathcal{S}_n$, therefore

$$\mathsf{len}(w^{(0)}, T) \geq d(w^{(0)}, \partial \mathcal{S}_n) \geq r_n(w^{(0)}). \tag{10}$$

Now we consider the following two cases.

*Case* (i). $\mathcal{L}_n(w^{(T)}) = \min_w \mathcal{L}_n(w)$. Since $\mathcal{L}_n(w^{(t)})$ is non-increasing, we have $\mathcal{L}_n(w^{(t)}) = \min_w \mathcal{L}_n(w)$ and $w^{(t)} = w^{(T)}$ for $t \geq T$. Notice that $\mathcal{S}_n$ is a closed set, thus $w^{(t)} \in \mathcal{S}_n$ for $t \in [0, T]$, meaning that $T = \infty$, which contradicts with $T < \infty$.

*Case* (ii). $\mathcal{L}_n(w^{(T)}) > \min_w \mathcal{L}_n(w)$. By chain rule, we have for all $t \in [0, T]$,

$$
\begin{aligned}
\frac{d\mathcal{L}_n(w^{(t)})}{dt} &= \left\langle \nabla \mathcal{L}_n(w^{(t)}), \frac{dw^{(t)}}{dt} \right\rangle \\
&= -\left\| \nabla \mathcal{L}_n(w^{(t)}) \right\| \left\| \frac{dw^{(t)}}{dt} \right\| \\
&\leq -c_n \left( \mathcal{L}_n(w^{(t)}) - \min_w \mathcal{L}(w) \right)^{\theta_n} \left\| \frac{dw^{(t)}}{dt} \right\|.
\end{aligned}
$$

Then we can bound the trajectory length $\mathsf{len}(w^{(0)}, T)$ as

$$
\begin{aligned}
\mathsf{len}(w^{(0)}, T) &= \int_0^T \left\| \frac{dw^{(t)}}{dt} \right\| dt \\
&\leq \int_0^T -\frac{1}{c_n} \left( \mathcal{L}_n(w^{(t)}) - \min_w \mathcal{L}(w) \right)^{-\theta_n} d\mathcal{L}_n(w^{(t)}) \\
&= \frac{1}{c_n(1-\theta_n)} \left[ \left( \mathcal{L}_n(w^{(0)}) - \min_w \mathcal{L}(w) \right)^{1-\theta_n} - \left( \mathcal{L}_n(w^{(T)}) - \min_w \mathcal{L}(w) \right)^{1-\theta_n} \right] \\
&< r_n(w^{(0)}),
\end{aligned}
\tag{11}
$$

which contradicts with (10).

$\square$

Now we begin to prove Theorem 1.

*Proof.* By Lemma B.1, we know that $\mathcal{L}_n(w^{(t)})$ satisfies Uniform-LGI for all $t \in [0, \infty)$. Then by the proof of Lemma B.1, we have $\forall t \in [0, \infty)$,

$$\mathsf{len}(w^{(0)}, t) \leq \frac{1}{c_n(1 - \theta_n)} \left[ \left( \mathcal{L}_n(w^{(0)}) - \min_w \mathcal{L}(w) \right)^{1-\theta_n} - \left( \mathcal{L}_n(w^{(t)}) - \min_w \mathcal{L}(w) \right)^{1-\theta_n} \right]. \tag{12}$$

For the convergence rate, note that

$$\begin{aligned} \frac{d \left( \mathcal{L}_n(w^{(t)}) - \min_w \mathcal{L}(w) \right)}{dt} &= \left\langle \nabla \mathcal{L}_n(w^{(t)}), \frac{dw^{(t)}}{dt} \right\rangle \\ &= - \left\| \nabla \mathcal{L}_n(w^{(t)}) \right\|^2 \\ &\leq -c_n^2 \left( \mathcal{L}_n(w^{(t)}) - \min_w \mathcal{L}(w) \right)^{2\theta_n}. \end{aligned} \tag{13}$$

Therefore

$$\left( \mathcal{L}_n(w^{(t)}) - \min_w \mathcal{L}(w) \right)^{-2\theta_n} d \left( \mathcal{L}_n(w^{(t)}) - \min_w \mathcal{L}(w) \right) \leq -c_n^2 dt.$$

Integrating on both sides of the equation, we can get $\forall t \in [0, \infty)$,

when $\theta_n = \frac{1}{2}$,

$$\mathcal{L}_n(w^{(t)}) - \min_w \mathcal{L}_n(w) \leq e^{-c_n^2 t} \left( \mathcal{L}_n(w^{(0)}) - \min_w \mathcal{L}_n(w) \right); \tag{14}$$

when $\frac{1}{2} < \theta_n < 1$,

$$\mathcal{L}_n(w^{(t)}) - \min_w \mathcal{L}_n(w) \leq (1 + Mt)^{-1/(2\theta_n - 1)} \left( \mathcal{L}_n(w^{(0)}) - \min_w \mathcal{L}_n(w) \right), \tag{15}$$

where $M = c_n^2 (2\theta_n - 1) \left( \mathcal{L}_n(w^{(0)}) - \min_w \mathcal{L}_n(w) \right)^{2\theta_n - 1}$.

Taking the limit on both sides of (14) and (15), since $w^{(\infty)}$ is the limit of $w^{(t)}$, we have

$$\mathcal{L}_n(w^{(\infty)}) = \min_w \mathcal{L}_n(w).$$

Hence $w^{(\infty)}$ is a global minimum. Taking the limit on both sides of (12), we get

$$\mathsf{len}(w^{(0)}, \infty) \leq r_n(w^{(0)}).$$

This completes the proof.

$\square$

## C    PROOF OF THEOREM 2

In this section, we will prove Theorem 2. This proof is based on the Rademacher complexity theory and the covering number of $\ell_2$ balls. Now we introduce some known technical lemmas that are used to build our proof.

The first lemma gives a generalization bound of a function class based on the Rademacher complexity, which is proved in (Mohri et al., 2018).

**Lemma C.1.** *Consider a family of functions $\mathcal{F}$ mapping from $\mathcal{Z}$ to $[a, b]$. Let $\mathcal{D}$ denote the distribution according to which samples are drawn. Then for any $\delta > 0$, with probability at least $1 - \delta$ over the draw of an i.i.d. sample $S = \{z_1, \dots, z_n\}$ of size $n$, the following holds for all $f \in \mathcal{F}$:*

$$\mathbb{E}_{z \sim \mathcal{D}} \left[ f(z) \right] - \frac{1}{n} \sum_{i=1}^{n} f(z_i) \leq 2\mathcal{R}_S(\mathcal{F}) + 3(b - a) \sqrt{\frac{\log(2/\delta)}{2n}},$$

where $\mathcal{R}_S(\mathcal{F})$ is the empirical Rademacher complexity with respect to the sample $S$, defined as:

$$\mathcal{R}_S(\mathcal{F}) = \mathbb{E}_\sigma \left[ \sup_{f \in \mathcal{F}} \frac{1}{n} \sum_{i=1}^n \sigma_i f(z_i) \right].$$

Here $\{\sigma_i\}_{i=1}^n$ are i.i.d. random variables drawn from $U\{-1, 1\}$.

In the next lemma, we prove a shifted version of the Ledoux-Talagrand contraction inequality (Ledoux & Talagrand, 2013), which is useful to bound the length-based Rademacher complexity.

**Lemma C.2.** *Let $g : \mathbb{R} \to \mathbb{R}$ be a convex and increasing function. Let $\phi_i : \mathbb{R} \to \mathbb{R}$ be $L$-Lipschitz functions, then for any bounded set $T \subset \mathbb{R}$ and any $t^{(0)} \in \mathbb{R}$, we have*

$$\mathbb{E}_\sigma \left[ g \left( \sup_{(t-t^{(0)}) \in T} \sum_{i=1}^n \sigma_i \left( \phi_i(t_i) - \phi_i \left( t_i^{(0)} \right) \right) \right) \right] \leq \mathbb{E}_\sigma \left[ g \left( L \sup_{(t-t^{(0)}) \in T} \sum_{i=1}^n \sigma_i \left( t_i - t_i^{(0)} \right) \right) \right],$$

*and*

$$\mathbb{E}_\sigma \left[ \sup_{(t-t^{(0)}) \in T} \left| \sum_{i=1}^n \sigma_i \left( \phi_i(t_i) - \phi_i \left( t_i^{(0)} \right) \right) \right| \right] \leq 2L \mathbb{E}_\sigma \left[ \sup_{(t-t^{(0)}) \in T} \left| \sum_{i=1}^n \sigma_i \left( t_i - t_i^{(0)} \right) \right| \right].$$

The special case for $t^{(0)} = 0$ corresponds to the original Ledoux-Talagrand contraction inequality. Here we prove a shifted version.

*Proof.* First notice that

$$\mathbb{E}_\sigma \left[ g \left( \sup_{(t-t^{(0)}) \in T} \sum_{i=1}^n \sigma_i \left( \phi_i(t_i) - \phi_i \left( t_i^{(0)} \right) \right) \right) \right] = \mathbb{E}_{\sigma_1,\ldots,\sigma_{n-1}} \left[ \mathbb{E}_{\sigma_n} \left[ g \left( \sup_{(t-t^{(0)}) \in T} \sum_{i=1}^n \sigma_i \left( \phi_i(t_i) - \phi_i \left( t_i^{(0)} \right) \right) \right) \right] \right].$$

Let $u_{n-1}(t) = \sum_{i=1}^{n-1} \sigma_i \left( \phi_i(t_i) - \phi_i \left( t_i^{(0)} \right) \right)$, then

$$\mathbb{E}_{\sigma_n} \left[ g \left( \sup_{(t-t^{(0)}) \in T} \sum_{i=1}^n \sigma_i \left( \phi_i(t_i) - \phi_i \left( t_i^{(0)} \right) \right) \right) \right]$$
$$= \frac{1}{2} g \left( \sup_{(t-t^{(0)}) \in T} u_{n-1}(t) + \left( \phi_n(t_n) - \phi_n \left( t_n^{(0)} \right) \right) \right) + \frac{1}{2} g \left( \sup_{(t-t^{(0)}) \in T} u_{n-1}(t) - \left( \phi_n(t_n) - \phi_n \left( t_n^{(0)} \right) \right) \right).$$

Suppose that the above two suprema can be reached at $t'$ and $\tilde{t}$ respectively, i.e.,

$$\sup_{(t-t^{(0)}) \in T} u_{n-1}(t) + \left( \phi_n(t_n) - \phi_n \left( t_n^{(0)} \right) \right) = u_{n-1}(t') + \left( \phi_n(t_n') - \phi_n \left( t_n^{(0)} \right) \right);$$
$$\sup_{(t-t^{(0)}) \in T} u_{n-1}(t) - \left( \phi_n(t_n) - \phi_n \left( t_n^{(0)} \right) \right) = u_{n-1}(\tilde{t}) - \left( \phi_n(\tilde{t}_n) - \phi_n \left( t_n^{(0)} \right) \right).$$

Otherwise we add an arbitrary positive number $\varepsilon$ in the above equations. Therefore,

$$\mathbb{E}_{\sigma_n} \left[ g \left( \sup_{(t-t^{(0)}) \in T} \sum_{i=1}^n \sigma_i \left( \phi_i(t_i) - \phi_i \left( t_i^{(0)} \right) \right) \right) \right]$$
$$= \frac{1}{2} \left[ g \left( u_{n-1}(t') + \left( \phi_n(t_n') - \phi_n \left( t_n^{(0)} \right) \right) \right) \right] + \frac{1}{2} \left[ g \left( u_{n-1}(\tilde{t}) - \left( \phi_n(\tilde{t}_n) - \phi_n \left( t_n^{(0)} \right) \right) \right) \right].$$

Without loss of generality, we assume

$$u_{n-1}(t') + \left(\phi_n(t'_n) - \phi_n\left(t_n^{(0)}\right)\right) \geq u_{n-1}(\tilde{t}) + \left(\phi_n(\tilde{t}_n) - \phi_n\left(t_n^{(0)}\right)\right);$$
$$u_{n-1}(\tilde{t}) - \left(\phi_n(\tilde{t}_n) - \phi_n\left(t_n^{(0)}\right)\right) \geq u_{n-1}(t') - \left(\phi_n(t'_n) - \phi_n\left(t_n^{(0)}\right)\right). \tag{16}$$

For the other cases, the method remains the same. We set

$$a = u_{n-1}(\tilde{t}) - \left(\phi_n(\tilde{t}_n) - \phi_n\left(t_n^{(0)}\right)\right),$$
$$b = u_{n-1}(\tilde{t}) - L\left(\tilde{t}_n - t_n^{(0)}\right),$$
$$a' = u_{n-1}(t') + L\left(t'_n - t_n^{(0)}\right),$$
$$b' = u_{n-1}(t') + \left(\phi_n(t'_n) - \phi_n\left(t_n^{(0)}\right)\right).$$

Now our goal is to prove:

$$g(a) - g(b) \leq g(a') - g(b'). \tag{17}$$

Considering the following four cases:

1. $t'_n \geq t_n^{(0)}$ and $\tilde{t}_n \geq t_n^{(0)}$. By the Lipschitzness of $\phi_n$ and equation (16) we know $a \geq b, b' \geq b$, and

$$(a - b) - (a' - b') = \phi_n(t'_n) - \phi_n(\tilde{t}_n) - L\left(t'_n - \tilde{t}_n\right).$$

If $t'_n \geq \tilde{t}_n$, we can get $a - b \leq a' - b'$. By the fact that $g$ is convex and increasing, we have $g(y + x) - g(x)$ is increasing in $y$ for every $x \geq 0$. Hence for $x = a - b$,

$$g(a) - g(b) = g(b + x) - g(b) \leq g(b' + x) - g(b') \leq g(a') - g(b').$$

If $t'_n < \tilde{t}_n$, we change $\phi_n$ into $-\phi_n$ and switch $t'$ and $\tilde{t}$, and the proof is similar.

2. $t'_n \leq t_n^{(0)}$ and $\tilde{t}_n \leq t_n^{(0)}$. Similarly, by changing the signs we can get the same result.

3. $t'_n \geq t_n^{(0)}$ and $\tilde{t}_n \leq t_n^{(0)}$. For this case we have $a \leq b$ and $b' \leq a'$, so $g(a) + g(b') \leq g(b) + g(a')$.

4. $t'_n \leq t_n^{(0)}$ and $\tilde{t}_n \geq t_n^{(0)}$. For this case we can change $\phi_n$ to $-\phi_n$, then we have $a \geq b$ and $a' \leq b'$, and finally we get $g(a) + g(b') \leq g(b) + g(a')$.

Thus equation (17) yields that

$$\mathbb{E}_{\sigma_n}\left[g\left(\sup_{(t-t^{(0)})\in T}\sum_{i=1}^n \sigma_i\left(\phi_i(t_i) - \phi_i\left(t_i^{(0)}\right)\right)\right)\right]$$
$$=\frac{1}{2}\left[g\left(u_{n-1}(t') + \left(\phi_n(t'_n) - \phi_n\left(t_n^{(0)}\right)\right)\right)\right] + \frac{1}{2}\left[g\left(u_{n-1}(\tilde{t}) - \left(\phi_n(\tilde{t}_n) - \phi_n\left(t_n^{(0)}\right)\right)\right)\right]$$
$$\leq\frac{1}{2}\left[g\left(u_{n-1}(t') + L\left(t'_n - t_n^{(0)}\right)\right)\right] + \frac{1}{2}\left[g\left(u_{n-1}(\tilde{t}) - L\left(\tilde{t}_n - t_n^{(0)}\right)\right)\right]$$
$$\leq\frac{1}{2}\left[g\left(\sup_{(t-t^{(0)})\in T}u_{n-1}(t) + L\left(t_n - t_n^{(0)}\right)\right)\right] + \frac{1}{2}\left[g\left(\sup_{(t-t^{(0)})\in T}u_{n-1}(t) - L\left(t_n - t_n^{(0)}\right)\right)\right]$$
$$=\mathbb{E}_{\sigma_n}\left[g\left(\sup_{(t-t^{(0)})\in T}u_{n-1}(t) + \sigma_n L\left(t_n - t_n^{(0)}\right)\right)\right]$$

Applying the same method to $\sigma_{n-1}, \ldots, \sigma_1$ successively, we obtain the first inequality

$$\mathbb{E}_\sigma \left[ g \left( \sup_{\left(t - t^{(0)}\right) \in T} \sum_{i=1}^{n} \sigma_i \left( \phi_i(t_i) - \phi_i \left(t_i^{(0)}\right) \right) \right) \right] \leq \mathbb{E}_\sigma \left[ g \left( L \sup_{\left(t - t^{(0)}\right) \in T} \sum_{i=1}^{n} \sigma_i \left( t_i - t_i^{(0)} \right) \right) \right].$$

For the second inequality, since $|x| = [x]_+ + [x]_-$ with $[x]_+ = \max(0, x)$ and $[x]_- = \max(0, -x)$,

$$\mathbb{E}_\sigma \left[ \sup_{\left(t - t^{(0)}\right) \in T} \left| \sum_{i=1}^{n} \sigma_i \left( \phi_i(t_i) - \phi_i \left(t_i^{(0)}\right) \right) \right| \right]$$

$$\leq \mathbb{E}_\sigma \left[ \sup_{\left(t - t^{(0)}\right) \in T} \left[ \sum_{i=1}^{n} \sigma_i \left( \phi_i(t_i) - \phi_i \left(t_i^{(0)}\right) \right) \right]_+ \right] + \mathbb{E}_\sigma \left[ \sup_{\left(t - t^{(0)}\right) \in T} \left[ \sum_{i=1}^{n} \sigma_i \left( \phi_i(t_i) - \phi_i \left(t_i^{(0)}\right) \right) \right]_- \right]$$

$$= 2\mathbb{E}_\sigma \left[ \sup_{\left(t - t^{(0)}\right) \in T} \left[ \sum_{i=1}^{n} \sigma_i \left( \phi_i(t_i) - \phi_i \left(t_i^{(0)}\right) \right) \right]_+ \right],$$

where the last equality is by $[-x]_- = [x]_+$ and $\sigma$ has the same distribution with $-\sigma$.

A simple fact is that

$$\sup_{\left(t - t^{(0)}\right) \in T} \left[ \sum_{i=1}^{n} \sigma_i \left( \phi_i(t_i) - \phi_i \left(t_i^{(0)}\right) \right) \right]_+ = \left[ \sup_{\left(t - t^{(0)}\right) \in T} \sum_{i=1}^{n} \sigma_i \left( \phi_i(t_i) - \phi_i \left(t_i^{(0)}\right) \right) \right]_+.$$

Since $\max(0, x)$ is convex and increasing, then by the first inequality we have

$$\mathbb{E}_\sigma \left[ \sup_{\left(t - t^{(0)}\right) \in T} \left[ \sum_{i=1}^{n} \sigma_i \left( \phi_i(t_i) - \phi_i \left(t_i^{(0)}\right) \right) \right]_+ \right] = \mathbb{E}_\sigma \left[ \left[ \sup_{\left(t - t^{(0)}\right) \in T} \sum_{i=1}^{n} \sigma_i \left( \phi_i(t_i) - \phi_i \left(t_i^{(0)}\right) \right) \right]_+ \right]$$

$$\leq \mathbb{E}_\sigma \left[ \left[ L \sup_{\left(t - t^{(0)}\right) \in T} \sum_{i=1}^{n} \sigma_i \left( t_i - t_i^{(0)} \right) \right]_+ \right]$$

$$\leq L\mathbb{E}_\sigma \left[ \sup_{\left(t - t^{(0)}\right) \in T} \left| \sum_{i=1}^{n} \sigma_i \left( t_i - t_i^{(0)} \right) \right| \right]$$

This completes the proof. $\qquad \square$

Now we apply Lemma C.2 to bound the empirical Rademacher complexity of an element-wise distance constrained function class. In the following lemma, all the notations are consistent with Theorem 2 unless stated otherwise.

**Lemma C.3.** *Given a function class $\mathcal{F}_{a,b} := \{x \mapsto f(w, x) : w \in \mathcal{S}_{a,b}\}$ and sample $S = \{x_1, \ldots, x_n\}$ with $\|x_i\| = 1$ for all $i \in [n]$, then we have*

$$\mathcal{R}_S(\mathcal{F}_{a,b}) \leq \sqrt{\frac{\|a\|^2 + \|b\|^2}{n}} \|L_\Psi(\mathcal{S}_{a,b})\|.$$

*Proof.* By definition,

$$
\begin{aligned}
n\mathcal{R}_S(\mathcal{F}_{a,b}) &= \mathbb{E}_\sigma\left[\sup_{w\in\mathcal{S}_{a,b}}\sum_{i=1}^n \sigma_i f(w,x_i)\right] \\
&= \mathbb{E}_\sigma\left[\sup_{w\in\mathcal{S}_{a,b}}\sum_{i=1}^n \sigma_i f(w,x_i)\right] - \mathbb{E}_\sigma\left[\sum_{i=1}^n \sigma_i f(0,x_i)\right] \\
&= \mathbb{E}_\sigma\left[\sup_{w\in\mathcal{S}_{a,b}}\sum_{i=1}^n \sigma_i\left(f(w,x_i)-f(0,x_i)\right)\right].
\end{aligned}
$$

Now we decompose the term $f(w,x_i)-f(0,x_i)$ as:

$$
\begin{aligned}
& f(w,x_i) - f(0,x_i) \\
={}& \Psi\left(x_i^\top\alpha_1,\ldots,x_i^\top\alpha_p,\beta_1,\ldots,\beta_q\right) - \Psi\left(0,\ldots,0,0,\ldots,0\right) \\
={}& \left(\Psi\left(x_i^\top\alpha_1,\ldots,x_i^\top\alpha_p,\beta_1,\ldots,\beta_q\right)-\Psi\left(0,\ldots,x_i^\top\alpha_p,\beta_1,\ldots,\beta_q\right)\right)+\left(\Psi\left(0,\ldots,x_i^\top\alpha_p,\beta_1,\ldots,\beta_q\right)-\Psi\left(0,0,\ldots,x_i^\top\alpha_p,\beta_1,\ldots,\beta_q\right)\right) \\
& + \cdots + \left(\Psi\left(0,\ldots,0,0,\ldots,0,\beta_q\right) - \Psi\left(0,\ldots,0,0,\ldots,0\right)\right).
\end{aligned}
$$

Then by the above decomposition and Lemma C.2, we have

$$
\begin{aligned}
& \mathbb{E}_\sigma\left[\sup_{w\in\mathcal{S}_{a,b}}\sum_{i=1}^n \sigma_i\left(f(w,x_i)-f(0,x_i)\right)\right] \\
\leq{}& L_\Psi^{(1)}(\mathcal{S}_{a,b})\mathbb{E}_\sigma\left[\sup_{w\in\mathcal{S}_{a,b}}\sum_{i=1}^n \sigma_i x_i^\top\alpha_1\right] + \cdots + L_\Psi^{(p+q)}(\mathcal{S}_{a,b})\mathbb{E}_\sigma\left[\sup_{w\in\mathcal{S}_{a,b}}\sum_{i=1}^n \sigma_i\beta_q\right].
\end{aligned}
$$

Notice that

$$
\begin{aligned}
\mathbb{E}_\sigma\left[\sup_{w\in\mathcal{S}_{a,b}}\sum_{i=1}^n \sigma_i x_i^\top\alpha_1\right] &= \mathbb{E}_\sigma\left[\sup_{\|\alpha_1\|\leq a_1}\sum_{i=1}^n \sigma_i x_i^\top\alpha_1\right] \\
&\leq a_1\mathbb{E}_\sigma\left[\left\|\sum_{i=1}^n \sigma_i x_i\right\|\right] \\
&\leq a_1\sqrt{\mathbb{E}_\sigma\left[\left\|\sum_{i=1}^n \sigma_i x_i\right\|^2\right]} \\
&= a_1\sqrt{n}.
\end{aligned}
$$

And

$$
\begin{aligned}
\mathbb{E}_\sigma\left[\sup_{w\in\mathcal{S}_{a,b}}\sum_{i=1}^n \sigma_i\beta_q\right] &= \mathbb{E}_\sigma\left[\sup_{|\beta_q|\leq b_q}\sum_{i=1}^n \sigma_i\beta_q\right] \\
&\leq b_q\mathbb{E}_\sigma\left[\left|\sum_{i=1}^n \sigma_i\right|\right] \\
&\leq b_q\sqrt{\mathbb{E}_\sigma\left[\left|\sum_{i=1}^n \sigma_i\right|^2\right]} \\
&= b_q\sqrt{n}.
\end{aligned}
$$

Therefore, by the Cauchy-Schwarz inequality, we can get

$$
\begin{aligned}
\mathbb{E}_\sigma\left[\sup_{w\in\mathcal{S}_{a,b}}\sum_{i=1}^n \sigma_i\left(f(w,x_i)-f(0,x_i)\right)\right] &\leq L_\Psi^{(1)}(\mathcal{S}_{a,b})a_1\sqrt{n} + \cdots + L_\Psi^{(p+q)}(\mathcal{S}_{a,b})b_q\sqrt{n} \\
&\leq \sqrt{n\left(\|a\|^2+\|b\|^2\right)}\,\|L_\Psi(\mathcal{S}_{a,b})\|.
\end{aligned}
$$

Finally, we have

$$\mathcal{R}_S(\mathcal{F}_{a,b}) \leq \sqrt{\frac{\|a\|^2 + \|b\|^2}{n}} \|L_\Psi(\mathcal{S}_{a,b})\|.$$

$\square$

Lemma C.3 gives an upper bound of the Rademacher complexity based on the element-wise distance. Notice that $\|w^{(\infty)}\| \leq \|w^{(0)}\| + \mathsf{len}(w^{(0)}, \infty)$. To obtain a length-based generalization bound, we consider to use $\mathcal{S}_{a,b}$ to cover the length-constrained space $\{w : \|w\| \leq R\}$, and then taking a union bound. For the $\ell_2$ ball covering number, we use the following result from (Neyshabur et al., 2019, Lemma 11).

**Lemma C.4.** *Given any* $\epsilon, D, \beta > 0$, *consider the set* $S_\beta^D = \{x \in \mathbb{R}^D : \|x\| \leq \beta\}$. *Then there exist* $N$ *sets* $\{T_i\}_{i=1}^N$ *of the form* $T_i = \{x \in \mathbb{R}^D : |x_j| \leq \alpha_j^i, \forall j \in [D]\}$ *such that* $S_\beta^D \subseteq \bigcup_{i=1}^N T_i$ *and* $\|\alpha^i\| \leq \beta(1 + \epsilon), \forall i \in [N]$ *where* $N = \binom{K+D-1}{D-1}$ *and*

$$K = \left\lceil \frac{D}{(1 + \epsilon)^2 - 1} \right\rceil.$$

**Lemma C.5.** *For any two positive integers* $n, k$ *with* $n \geq k$, *we have*

$$\binom{n}{k} \leq \left(\frac{en}{k}\right)^k.$$

*Proof.* Note that

$$\binom{n}{k} = \frac{n!}{k!(n-k)!} \leq \frac{n^k}{k!} \leq e^k \left(\frac{n}{k}\right)^k.$$

The last step is by

$$e^k = \sum_{i=0}^\infty \frac{k^i}{i!} \geq \frac{k^k}{k!}.$$

$\square$

Now combining Lemma C.1, C.3, C.4 and C.5, we are ready to prove Theorem 2.

*Proof of Theorem 2.* First we apply Lemma C.4 with $\epsilon = \sqrt{2} - 1$, $D = p + q$, and $\beta = M_\delta + R_{n,\delta}$, then there exist $N$ sets $\mathcal{S}_{a^k,b^k}$ such that $S_\beta^D \subseteq \bigcup_{k=1}^N \mathcal{S}_{a^k,b^k}$ and $\sqrt{\|a^k\|^2 + \|b^k\|^2} \leq \sqrt{2}\beta$, with $N = \binom{2D-1}{D-1}$.

Therefore, for each parameter space $\mathcal{S}_{a^k,b^k}$, by Lemma C.3 we have

$$\mathcal{R}_S(\mathcal{F}_{a^k,b^k}) \leq \sqrt{\frac{2}{n}}\beta \left\|L_\Psi(\mathcal{S}_{a^k,b^k})\right\|.$$

Notice that the local Lipschitz constant of $\ell$ in $\mathcal{S}_{a^k,b^k}$ is $L_\ell(\mathcal{S}_{a^k,b^k})$. Hence, by Lemma C.1 and the Ledoux-Talagrand contraction inequality, for any $\delta > 0$, with probability at least $1 - \delta/N$ over the training sample, the following holds for all $w \in \mathcal{S}_{a^k,b^k}$:

$$\mathcal{L}_\mathcal{D}(w) \leq \mathcal{L}_n(w) + \frac{2\sqrt{2}\beta L_\ell(\mathcal{S}_{a^k,b^k}) \left\|L_\Psi(\mathcal{S}_{a^k,b^k})\right\|}{\sqrt{n}} + 3M_\beta \sqrt{\frac{\log(2N/\delta)}{2n}},$$

where $M_\beta = \sup_{\|a^k\|^2 + \|b^k\|^2 \leq 2\beta^2} \sup_{w \in \mathcal{S}_{a^k,b^k}, \|x\| \leq 1, |y| \leq 1} \ell(f(w,x), y)$.

Since $w^{(\infty)} \in S_\beta^D \subseteq \bigcup_{k=1}^N \mathcal{S}_{a^k,b^k}$, by taking the union bound over all sets $\mathcal{S}_{a^k,b^k}$, we can get with probability at least $1 - \delta$ over the initialization $\mathcal{I}$ the training sample,

$$\mathcal{L}_\mathcal{D}(w^{(\infty)}) \leq \mathcal{L}_n(w^{(\infty)}) + \sup_{\|a\|^2 + \|b\|^2 \leq 2\beta^2} \frac{2\sqrt{2}\beta L_\ell(\mathcal{S}_{a,b}) \left\|L_\Psi(\mathcal{S}_{a,b})\right\|}{\sqrt{n}} + 3M_{a,b} \sqrt{\frac{\log(2N/\delta)}{2n}}.$$

Theorem 1 already showed that $\mathcal{L}_n(w^{(\infty)}) = \min_w \mathcal{L}_n(w)$. Thus it remains to bound the term $\log N$. For $D = 1$, $N = 1$. For $D \geq 2$, by Lemma C.5,

$$\log N \leq (D-1) \log \left( \frac{e(2D-1)}{D-1} \right) < 2.1(D-1) < 3D = 3(p+q).$$

Finally, let $R = \sqrt{2}\beta$, we complete the proof of Theorem 2.

$\square$

### C.1 ADDITIONAL OF RESULTS FOR THE GENERALIZATION DURING TRAINING THE MODEL

Theorem 2 gives a length-based generalization bound for the final model. In this subsection, we apply our framework to derive generalization estimates that evolves according to the length of time (number of epochs) of training by combining the length estimate obtained in Theorem 1.

The approach is to give a generalization bound for *early stopping* when the loss value first reaches $\varepsilon \geq 0$. The idea is that, when there exists $T > 0$ such that $\mathcal{L}_n(w^T) = \varepsilon$, then by the inequality (11) in the proof of Lemma B.1, we can get an upper bound for the length $\mathsf{len}(w^{(0)}, T)$ in terms of $\mathcal{L}_n(w^{(0)}), \min_w \mathcal{L}(w), \varepsilon, c_n, \theta_n$. Finally, we can get a generalization bound by our new length estimate.

To get a clean expression for the generalization bound, we assume the optimal value of the empirical loss function $\mathcal{L}_n(w)$ is zero. Then the rigorous statement is stated as follows:

**Corollary 1.** *Consider a training criterion of early stopping that the training is stopped once the empirical loss value first reaches $\varepsilon \geq 0$. Then for any given $\varepsilon \geq 0$, under the notations and conditions in Theorem 1, suppose that for any $\delta \in (0,1)$, there exists $r_{n,\delta}$ such that $r_n(w^{(0)}) \leq r_{n,\delta}$ with probability at least $1 - \delta$ over the initialization and the training samples. Then, we have with probability at least $1 - \delta$ over initialization and the training samples, the generalization error for the stopping parameter $w$ is given by:*

$$\mathcal{L}_{\mathcal{D}}(w) \leq \varepsilon + \sup_{\|a\|^2 + \|b\|^2 \leq r_{\varepsilon,n,\delta}^2} \frac{2r_{\varepsilon,n,\delta} L_\ell(\mathcal{S}_{a,b}) \|L_\Psi(\mathcal{S}_{a,b})\|}{\sqrt{n}} + 3M_{a,b}\sqrt{\frac{3(p+q) + \log(2/\delta)}{2n}}, \quad (18)$$

*where $r_{\varepsilon,n,\delta} = \sqrt{2} \left( M_\delta + r_{n,\delta} - \frac{\varepsilon^{1-\theta_n}}{c_n(1-\theta_n)} \right)$.*

**Remark 2.** *Corollary 1 shows a trade-off between $\varepsilon$ and the term $r_{\varepsilon,n,\delta}$. The case for $\varepsilon = 0$ corresponds to the combining results of Theorem 2 and Theorem 1.*

*Proof.* The proof is straightforward. Notice that by the inequality (11), we can bound the length $\mathsf{len}(w^{(0)}, w)$ as

$$\mathsf{len}(w^{(0)}, w) \leq r_n(w^{(0)}) - \frac{\varepsilon^{1-\theta_n}}{c_n(1-\theta_n)},$$

where $r_n(w^{(0)})$ is specified in equation (2). Then by the same argument in the proof of Theorem 2, we may directly replace $r_n(w^{(0)})$ with $r_n(w^{(0)}) - \frac{\varepsilon^{1-\theta_n}}{c_n(1-\theta_n)}$ to get the desired bound. $\square$

## D PROOFS FOR SECTION 4

In this section, our goal is to prove all the theorems in Section 4. A crucial part of the proofs is the spectral analysis of the random matrix $\mathcal{X}$. Therefore, we start with introducing the non-asymptotic results of $\lambda_{\max}(\mathcal{X}\mathcal{X}^\top)$ and $\lambda_{\min}(\mathcal{X}\mathcal{X}^\top)$ from (Rudelson & Vershynin, 2010).

The first result is from (Rudelson & Vershynin, 2010, Proposition 2.4), characterizing the non-asymptotic behavior of the largest singular value of subgaussian matrices.

**Lemma D.1.** *Let $A$ be an $N \times n$ random matrix whose entries are independent mean zero subgaussian random variables whose subgaussian moments are bounded by $1$. Then for every $t \geq 0$, with probability at least $1 - 2e^{-ct^2}$ over the randomness of the entries,*

$$\sqrt{\lambda_{\max}(AA^\top)} \leq C(\sqrt{N} + \sqrt{n}) + t,$$

*where $c$ and $C$ are two positive constants that depend only on the subgaussian moment of the entries.*

The second result is from (Rudelson & Vershynin, 2009, Theorem 1.1), characterizing the non-asymptotic behavior of the smallest singular value of subgaussian matrices.

**Lemma D.2.** *Let $A$ be an $N \times n$ random matrix whose entries are independent and identically distributed subgaussian random variables with zero mean and unit variance. If $N > n$, then for every $\varepsilon > 0$, with probability at least $1 - (C_1 \varepsilon)^{N-n+1} - c_1^N$ over the randomness of the entries,*

$$\sqrt{\lambda_{\min}(A^\top A)} \geq \varepsilon(\sqrt{N} - \sqrt{n-1}),$$

*where $C_1 > 0$ and $c_1 \in (0,1)$ depend only on the subgaussian moment of the entries.*

### D.1 PROOF OF THEOREM 3

In this section, we will prove Theorem 3 based on the three steps in our framework. All the notations are consistent with Theorem 3 unless stated otherwise.

*Proof of Theorem 3.* First, we prove the result for Step 1.

For a vector $a = (a_1, \ldots, a_n)^\top \in \mathbb{R}^n$, we use $a^{\circ m}$ to denote the element-wise power, i.e., $a^{\circ m} = (a_1^m, \ldots, a_n^m)^\top$. For the $\ell_p$ linear regression loss function $\mathcal{L}_n(w)$, notice that

$$\nabla \mathcal{L}_n(w) = \frac{1}{n} \sum_{i=1}^n \left(w^\top x_i - y_i\right)^{p-1} x_i = \frac{1}{n} \mathcal{X}^\top \left(\mathcal{X}w - \mathcal{Y}\right)^{\circ(p-1)}.$$

Then since $\mathcal{X}$ has full row rank, we have $\forall w \in \mathbb{R}^d$,

$$\begin{aligned}
\|\nabla \mathcal{L}_n(w)\| &= \frac{1}{n} \left\| \mathcal{X}^\top \left(\mathcal{X}w - \mathcal{Y}\right)^{\circ(p-1)} \right\| \\
&\geq \frac{\sqrt{\lambda_{\min}(\mathcal{X}\mathcal{X}^\top)}}{n} \left\| \left(\mathcal{X}w - \mathcal{Y}\right)^{\circ(p-1)} \right\| \\
&= \frac{\sqrt{\lambda_{\min}(\mathcal{X}\mathcal{X}^\top)}}{n} \|\mathcal{X}w - \mathcal{Y}\|_{2p-2}^{p-1} \\
&\geq \frac{\sqrt{\lambda_{\min}(\mathcal{X}\mathcal{X}^\top)}}{n} \|\mathcal{X}w - \mathcal{Y}\|_p^{p-1} \cdot n^{1/p - 1/2} \\
&= p^{1-1/p} \sqrt{\frac{\lambda_{\min}(\mathcal{X}\mathcal{X}^\top)}{n}} \mathcal{L}_n(w)^{1-1/p}.
\end{aligned}$$

Therefore,

$$c_n = p^{1-1/p} \sqrt{\frac{\lambda_{\min}(\mathcal{X}\mathcal{X}^\top)}{n}}, \quad \theta_n = 1 - 1/p.$$

For Step 2, note that $\min_w \mathcal{L}_n(w) = 0$, then the result can be proved by directly plugging $c_n$ and $\theta_n$ into Theorem 1.

Next, we prove the result for Step 3. By Theorem 1 and the property of the target function, we have for any $w^{(0)}$ that satisfies $\left\|w^{(0)}\right\|_2 \leq c_0$,

$$
\begin{aligned}
r_n(w^{(0)}) &= \frac{\sqrt{n}\left(p\mathcal{L}_n(w^{(0)})\right)^{1/p}}{\sqrt{\lambda_{\min}(\mathcal{X}\mathcal{X}^\top)}} \\
&= n^{1/2-1/p}\frac{\left\|\mathcal{X}w^{(0)} - \mathcal{Y}\right\|_p}{\sqrt{\lambda_{\min}(\mathcal{X}\mathcal{X}^\top)}} \\
&\leq n^{1/2-1/p}\frac{\left\|\mathcal{X}w^{(0)} - \mathcal{Y}\right\|_2}{\sqrt{\lambda_{\min}(\mathcal{X}\mathcal{X}^\top)}} \\
&\leq n^{1/2-1/p}\sqrt{\frac{\lambda_{\max}(\mathcal{X}\mathcal{X}^\top)}{\lambda_{\min}(\mathcal{X}\mathcal{X}^\top)}}\,(c_0 + c^*).
\end{aligned}
$$

Now we apply Lemma D.1 with $A = \mathcal{X}$ and $t = \sqrt{\frac{\log(4/\delta)}{c}}$, then we have with probability at least $1 - \delta/2$ over the samples,

$$
\sqrt{\lambda_{\max}(\mathcal{X}\mathcal{X}^\top)} \leq C(\sqrt{n} + \sqrt{d}) + \sqrt{\frac{\log(4/\delta)}{c}}, \tag{19}
$$

where $c$ and $C$ are two positive constants that depend only on the subgaussian moment of the entries.

Similarly, let $\tau = c_1 \in (0, 1), \varepsilon = \tau/C_1 > 0$, then Lemma D.2 implies that with probability at least $1 - \tau^{d-n+1} - \tau^d$ over the samples,

$$
\sqrt{\lambda_{\min}(\mathcal{X}\mathcal{X}^\top)} \geq \frac{\tau}{C_1}(\sqrt{d} - \sqrt{n-1}), \tag{20}
$$

where $C_1 > 0$ and $\tau \in (0, 1)$ depend only on the subgaussian moment of the entries.

Taking the union bound, we have with probability at least $1 - \delta/2 - \tau^{d-n+1} - \tau^d$ over the initialization and the training samples,

$$
\begin{aligned}
\frac{r_n(w^{(0)})}{\sqrt{n}} &\leq n^{-1/p}(c_0 + c^*)\frac{C(\sqrt{n} + \sqrt{d}) + \sqrt{\frac{\log(4/\delta)}{c}}}{\frac{\tau}{C_1}(\sqrt{d} - \sqrt{n-1})} \\
&\leq n^{-1/p}(c_0 + c^*)\frac{C\left(\sqrt{\frac{n}{d}} + 1\right) + \sqrt{\frac{\log(4/\delta)}{cd}}}{\frac{\tau}{C_1}\left(1 - \sqrt{\frac{n-1}{d}}\right)} \\
&\leq n^{-1/p}(c_0 + c^*)\frac{CC_1(\sqrt{\gamma_1} + 1) + C_1\sqrt{\frac{\gamma_1\log(4/\delta)}{cn}}}{\tau(1 - \sqrt{\gamma_1})} \\
&\leq \mathcal{O}\left(n^{-1/p} + n^{-1/p}\sqrt{\frac{\log(1/\delta)}{n}}\right).
\end{aligned} \tag{21}
$$

Recall for the linear regression model (5), $f(w, x) = w^\top x$. Thus $\Psi(x) = x$ is an identity function with $p = 1, q = 0$, and $\|L_\Psi(\mathcal{S}_{a,b})\| = 1$ for any $a, b$. Since the loss function $\tilde{\ell}$ is bounded by 1 and 1-Lipschitz, we know that $L_\ell(\mathcal{S}_{a,b}) = M_R = 1$ for any $a$ and $b$. Finally by Theorem 2 and $\tilde{\ell}(y, y) = 0$, we have with probability at least $1 - \delta/2$ over the samples,

$$
\mathbb{E}_{(x,y)\sim\mathcal{D}}\left[\tilde{\ell}\left(f(w^{(\infty)}, x), y\right)\right] \leq \frac{2\sqrt{2}(r_n(w^{(0)}) + c_0)}{\sqrt{n}} + 3\sqrt{\frac{3 + \log(4/\delta)}{2n}}.
$$

Combining the inequality (21), we have with probability at least $1 - \delta - \tau^{d-n+1} - \tau^d$ over training samples,

$$
\mathbb{E}_{(x,y)\sim\mathcal{D}}\left[\tilde{\ell}\left(f(w^{(\infty)}, x), y\right)\right] \leq \mathcal{O}\left(n^{-1/p}\right) + \mathcal{O}\left(\sqrt{\frac{\log(1/\delta)}{n}}\right).
$$

This completes the proof of Theorem 3.

$\square$

## D.2 PROOF OF THEOREM 4

In this section, we will prove Theorem 4. First, we present some useful lemmas for proving our results, and then we give the proofs of Theorem 4 for the RBF kernel and the inner product kernel separately.

For the RBF kernel $k(x, y) = \varrho(\|y - x\|)$, the following two lemmas give non-asymptotic bounds for $\lambda_{\max}(k(\mathcal{X}, \mathcal{X}))$ and $\lambda_{\min}(k(\mathcal{X}, \mathcal{X}))$ based on the separation distance $\mathsf{SD}$ of $\mathcal{X}$.

The first lemma is from (Diederichs & Iske, 2019, Lemma 3.1), providing an upper bound for $\lambda_{\max}(k(\mathcal{X}, \mathcal{X}))$.

**Lemma D.3.** *For the RBF kernel, if $\varrho : \mathbb{R}_{\geq 0} \to \mathbb{R}_{\geq 0}$ is a decreasing function, then*

$$\lambda_{\max}(k(\mathcal{X}, \mathcal{X})) \leq \varrho(0) + 3d \sum_{t=1}^{\infty} (t + 2)^{d-1} \varrho(t \cdot \mathsf{SD}), \tag{22}$$

*and the sum of the infinite series in equation (22) is finite if and only if $\varrho(\|x\|) \in L^1(\mathbb{R}^d)$.*

The next lemma is from (Wendland, 2004, Theorem 12.3), giving a lower bound for $\lambda_{\min}(k(\mathcal{X}, \mathcal{X}))$.

**Lemma D.4.** *Suppose that $k$ is a positive-definite RBF kernel. If $\varrho(\|x\|) \in L^1(\mathbb{R}^d)$, one can define the Fourier transform of $\varrho$ as $\hat{\varrho}(\omega) := (2\pi)^{-d/2} \int_{\mathbb{R}^d} \varrho(\omega) e^{-ix^\top \omega} d\omega$. With a decreasing function $\varrho_0(M)$ and two constants $M_d, C_d$ defined as*

$$\varrho_0(M) := \inf_{\|x\| \leq 2M} \hat{\varrho}(x), \quad M_d = 6.38d, \quad C_d = \frac{1}{2\Gamma(d/2 + 1)} \left(\frac{M_d}{2^{3/2}}\right)^d,$$

*where $\Gamma$ is the gamma function. Then a lower bound on $\lambda_{\min}(k(\mathcal{X}, \mathcal{X}))$ is given by*

$$\lambda_{\min}(k(\mathcal{X}, \mathcal{X})) \geq C_d \cdot \varrho_0(M_d/\mathsf{SD}) \cdot \mathsf{SD}^{-d}.$$

For the inner product kernel $k(x, y) = \varrho\left(\frac{x^\top y}{d}\right)$, it is shown in (El Karoui et al., 2010) that the kernel matrix $k(\mathcal{X}, \mathcal{X})$ can be approximated by the linear combination of all-ones matrix $11^\top$, sample covariance matrix $\mathcal{X}\mathcal{X}^\top$ and identity matrix. To obtain non-asymptotic results on the spectra of the kernel matrix $k(\mathcal{X}, \mathcal{X})$, we borrow the technique from (Liang & Rakhlin, 2020, Proposition A.2), and show the result for subgaussian entries in the next lemma.

**Lemma D.5.** *For the inner product kernel, suppose that the entries of $\mathcal{X}$ are i.i.d. subgaussian random variables with zero mean and unit variance, then with probability at least $1 - \delta - d^{-2}$ over the entries,*

$$\left\|k(\mathcal{X}, \mathcal{X}) - k^{\lin}(\mathcal{X}, \mathcal{X})\right\| \leq d^{-1/2}\left(\delta^{-1/2} + \log^{0.51} d\right),$$

*where $k^{\lin}(\mathcal{X}, \mathcal{X})$ is defined as*

$$k^{\lin}(\mathcal{X}, \mathcal{X}) := \left(\varrho(0) + \frac{\varrho''(0)}{d}\right) 11^\top + \varrho'(0) \frac{\mathcal{X}\mathcal{X}^\top}{d} + (\varrho(1) - \varrho(0) - \varrho'(0)) \, \mathbb{I}_{n \times n}.$$

*Proof.* Note that the sample covariance matrix $\Sigma_d = \mathbb{I}_{d \times d}$, then by applying (Liang & Rakhlin, 2020, Proposition A.2) with subgaussian random entries we can prove this lemma.

$\square$

**Lemma D.6.** *Suppose that $A, B \in \mathbb{R}^{n \times n}$ are two symmetric matrices, then we have*

$$\lambda_{\min}(A + B) \geq \lambda_{\min}(A) + \lambda_{\min}(B).$$

*Proof.* Note that for any $x \in \mathbb{R}^n$ with $\|x\| = 1$,

$$x^\top (A + B)x = x^\top Ax + x^\top Bx \geq \lambda_{\min}(A) + \lambda_{\min}(B).$$

By definition, we have

$$\lambda_{\min}(A + B) = \inf_{\|x\|=1} x^\top (A + B)x \geq \lambda_{\min}(A) + \lambda_{\min}(B),$$

which completes the proof.

$\square$

Now we are ready to prove Theorem 4.

*Proof of Theorem 4.* For Step 1. Notice that $\forall w \in \mathbb{R}^s$,

$$\begin{aligned}
\|\nabla \mathcal{L}_n(w)\| &= \frac{1}{n} \left\| \varphi(\mathcal{X})^\top \left( \varphi(\mathcal{X})w - \mathcal{Y} \right)^{\circ(p-1)} \right\| \\
&\geq \frac{\sqrt{\lambda_{\min}(\varphi(\mathcal{X})\varphi(\mathcal{X})^\top)}}{n} \left\| (\varphi(\mathcal{X})w - \mathcal{Y})^{\circ(p-1)} \right\| \\
&= \frac{\sqrt{\lambda_{\min}(k(\mathcal{X}, \mathcal{X}))}}{n} \|\varphi(\mathcal{X})w - \mathcal{Y}\|_{2p-2}^{p-1} \\
&\geq \frac{\sqrt{\lambda_{\min}(k(\mathcal{X}, \mathcal{X}))}}{n} \|\varphi(\mathcal{X})w - \mathcal{Y}\|_p^{p-1} \cdot n^{1/p-1/2} \\
&= p^{1-1/p} \sqrt{\frac{\lambda_{\min}(k(\mathcal{X}, \mathcal{X}))}{n}} \mathcal{L}_n(w)^{1-1/p}.
\end{aligned}$$

Therefore, $\mathcal{L}_n(w)$ satisfies the Uniform-LGI globally on $\mathbb{R}^s$ with

$$c_n = p^{1-1/p} \sqrt{\frac{\lambda_{\min}(k(\mathcal{X}, \mathcal{X}))}{n}}, \quad \theta_n = 1 - 1/p.$$

For Step 2. Since $k$ is a positive-definite kernel, and $\theta_n = 1 - 1/p$, then $\min_w \mathcal{L}_n(w) = 0$, thus by Theorem 1 we can directly get the result.

The proof of Step 3 is two-sided. First, since $\forall x \in \mathcal{X}, \|\varphi(x)\| = \sqrt{k(x, x)} \leq 1$, then the kernel regression model (7) can be viewed as $\ell_p$ linear regression on inputs $\varphi(\mathcal{X})$. Hence, $\Psi$ is an identity function with $p = 1, q = 0$, and $\|L_\Psi(\mathcal{S}_{a,b})\| = L_\ell(\mathcal{S}_{a,b}) = M_R = 1$ for any $a, b$. This means that we only need to bound the term $r_n(w^{(0)})/\sqrt{n}$.

By Theorem 1 and the property of the target function, for any $w^{(0)}$ that satisfies $\left\| w^{(0)} \right\|_2 \leq c_0$, we have

$$\begin{aligned}
r_n(w^{(0)}) &= \frac{\sqrt{n} \left( p\mathcal{L}_n(w^{(0)}) \right)^{1/p}}{\sqrt{\lambda_{\min}(k(\mathcal{X}, \mathcal{X}))}} \\
&= n^{1/2-1/p} \frac{\left\| \varphi(\mathcal{X})w^{(0)} - \mathcal{Y} \right\|_p}{\sqrt{\lambda_{\min}(k(\mathcal{X}, \mathcal{X}))}} \\
&\leq n^{1/2-1/p} \frac{\left\| \varphi(\mathcal{X})w^{(0)} - \mathcal{Y} \right\|_2}{\sqrt{\lambda_{\min}(k(\mathcal{X}, \mathcal{X}))}} \\
&\leq n^{1/2-1/p} \frac{\left\| \varphi(\mathcal{X})w^{(0)} \right\| + \|\mathcal{Y}\|}{\sqrt{\lambda_{\min}(k(\mathcal{X}, \mathcal{X}))}} \\
&\leq n^{1/2-1/p} \sqrt{\frac{\lambda_{\max}(k(\mathcal{X}, \mathcal{X}))}{\lambda_{\min}(k(\mathcal{X}, \mathcal{X}))}} (c_0 + c^*).
\end{aligned} \tag{23}$$

Then for the RBF kernel, Lemma D.3 and Lemma D.4 indicate that there exists a positive constant $C(\varrho, d, q_{\min}, q_{\max})$ that only depends on $\varrho, d, q_{\min}, q_{\max}$ such that

$$\frac{\lambda_{\max}(k(\mathcal{X}, \mathcal{X}))}{\lambda_{\min}(k(\mathcal{X}, \mathcal{X}))} \leq C(\varrho, d, q_{\min}, q_{\max}), \quad \forall n \geq 1,$$

which implies that for all initialization $w^{(0)}$,

$$\frac{r_n(w^{(0)})}{\sqrt{n}} \leq \mathcal{O}\left(n^{-1/p}\right). \tag{24}$$

Therefore, by Theorem 2 and Lemma C.1, we have with probability at least $1 - \delta$ over training samples,

$$\mathbb{E}_{(x,y) \sim \mathcal{D}}\left[\tilde{\ell}\left(f(w^{(\infty)}, x), y\right)\right] \leq \mathcal{O}\left(n^{-1/p}\right) + \mathcal{O}\left(\sqrt{\frac{\log(1/\delta)}{n}}\right),$$

which completes the proof for the RBF kernel.

For the inner product kernel, to obtain an upper bound for $\frac{\lambda_{\max}(k(\mathcal{X}, \mathcal{X}))}{\lambda_{\min}(k(\mathcal{X}, \mathcal{X}))}$, first notice that

$$\begin{aligned}
\lambda_{\max}(k(\mathcal{X}, \mathcal{X})) &= \|k(\mathcal{X}, \mathcal{X})\| \\
&\leq \left\|k^{\lin}(\mathcal{X}, \mathcal{X})\right\| + \left\|k(\mathcal{X}, \mathcal{X}) - k^{\lin}(\mathcal{X}, \mathcal{X})\right\|.
\end{aligned} \tag{25}$$

By Lemma D.6, we can get

$$\begin{aligned}
\lambda_{\min}(k(\mathcal{X}, \mathcal{X})) &\geq \lambda_{\min}(k^{\lin}(\mathcal{X}, \mathcal{X})) + \lambda_{\min}(k(\mathcal{X}, \mathcal{X}) - k^{\lin}(\mathcal{X}, \mathcal{X})) \\
&\geq \lambda_{\min}(k^{\lin}(\mathcal{X}, \mathcal{X})) - \left\|k(\mathcal{X}, \mathcal{X}) - k^{\lin}(\mathcal{X}, \mathcal{X})\right\|.
\end{aligned} \tag{26}$$

Under Assumption 2, Lemma D.5 implies that

$$\begin{aligned}
\left\|k^{\lin}(\mathcal{X}, \mathcal{X})\right\| &\leq \frac{\varrho''(0)}{d}\left\|11^\top\right\| + \frac{\varrho'(0)}{d}\left\|\mathcal{X}\mathcal{X}^\top\right\| + (\varrho(1) - \varrho'(0)) \\
&\leq \frac{n\varrho''(0)}{d} + \varrho'(0)\frac{\lambda_{\max}(\mathcal{X}\mathcal{X}^\top)}{d} + (\varrho(1) - \varrho'(0)) \\
&\leq \gamma_1\varrho''(0) + \varrho'(0)\frac{\lambda_{\max}(\mathcal{X}\mathcal{X}^\top)}{d} + (\varrho(1) - \varrho'(0)),
\end{aligned}$$

and

$$\lambda_{\min}(k^{\lin}(\mathcal{X}, \mathcal{X})) \geq \varrho(1) - \varrho'(0) > 0.$$

Thus, by equation (26) we have

$$\lambda_{\min}(k(\mathcal{X}, \mathcal{X})) \geq (\varrho(1) - \varrho'(0)) - \left\|k(\mathcal{X}, \mathcal{X}) - k^{\lin}(\mathcal{X}, \mathcal{X})\right\|. \tag{27}$$

Under Assumption 1, by equation (19), we have with probability at least $1 - \delta/3$ over the samples,

$$\begin{aligned}
\frac{\lambda_{\max}(\mathcal{X}\mathcal{X}^\top)}{d} &\leq \left(C\left(\sqrt{\frac{n}{d}} + 1\right) + \sqrt{\frac{\log(6/\delta)}{cd}}\right)^2 \\
&\leq \left(C\left(\sqrt{\gamma_1} + 1\right) + \sqrt{\frac{\gamma_1 \log(6/\delta)}{cn}}\right)^2.
\end{aligned}$$

Therefore, by equation (25), with probability at least $1 - \delta/3$ over the samples,

$$\lambda_{\max}(k(\mathcal{X}, \mathcal{X})) \leq \gamma_1\varrho''(0) + \varrho'(0)\left(C\left(\sqrt{\gamma_1} + 1\right) + \sqrt{\frac{\gamma_1 \log(6/\delta)}{cn}}\right)^2 + \varrho(1) - \varrho'(0) + \left\|k(\mathcal{X}, \mathcal{X}) - k^{\lin}(\mathcal{X}, \mathcal{X})\right\|. \tag{28}$$

By Lemma D.5, for large $d$ and small $\delta$ such that $d^{-1/2}\left(\sqrt{3}\delta^{-1/2} + \log^{0.51} d\right) \leq 0.5(\varrho(1) - \varrho'(0))$, we have with probability at least $1 - \delta/3 - d^{-2}$ over the entries,

$$\left\|k(\mathcal{X}, \mathcal{X}) - k^{\text{lin}}(\mathcal{X}, \mathcal{X})\right\| \leq 0.5(\varrho(1) - \varrho'(0)).$$

Then equation (27) and (28) yields that with probability at least $1 - 2\delta/3 - d^{-2}$ over the samples,

$$\lambda_{\min}(k(\mathcal{X}, \mathcal{X})) \geq 0.5(\varrho(1) - \varrho'(0)),$$

$$\lambda_{\max}(k(\mathcal{X}, \mathcal{X})) \leq \gamma_1 \varrho''(0) + \varrho'(0)\left(C\left(\sqrt{\gamma_1} + 1\right) + \sqrt{\frac{\gamma_1 \log(4/\delta)}{cn}}\right)^2 + 1.5(\varrho(1) - \varrho'(0)).$$

Hence, by equation (23), with probability at least $1 - 2\delta/3 - d^{-2}$ over the samples, we have

$$\frac{r_n(w^{(0)})}{\sqrt{n}} \leq \mathcal{O}\left(n^{-1/p} + \frac{\sqrt{\log(1/\delta)}}{n^{1/2+1/p}}\right). \tag{29}$$

Combining Theorem 2, we get with probability at least $1 - \delta - d^{-2}$ over the samples,

$$\mathbb{E}_{(x,y)\sim\mathcal{D}}\left[\tilde{\ell}\left(f(w^{(\infty)}, x), y\right)\right] \leq \mathcal{O}\left(n^{-1/p}\right) + \mathcal{O}\left(\sqrt{\frac{\log(1/\delta)}{n}}\right),$$

which completes the proof.

$\square$

### D.3 PROOF OF THEOREM 5

In this section, we will prove Theorem 5. We first introduce some important lemmas for proving our final results. Lemma D.7 shows that the smallest eigenvalue of the NTK matrix $\Theta(t)$ has a lower bounded given the overparameterization, by which we can prove the optimization result. In Lemma D.8, we show that the eigenvalues of the NTK matrix are related to the data covariance matrix. Then by combining Lemma D.11 and Lemma D.9 we can prove the generalization result.

**Lemma D.7.** *For any* $\delta \in (0, 1)$*, if* $m \geq \text{poly}\left(n, \lambda_{\min}^{-1}(\widehat{\Theta}), \delta^{-1}\right)$*, then with probability at least* $1 - \delta$ *over the random initialization,*

$$\lambda_{\min}(\Theta(t)) \geq \frac{1}{2}\lambda_{\min}(\widehat{\Theta}), \quad \forall t \geq 0.$$

*Proof.* The proof is the same as the proof of (Du et al., 2019, Lemma 3.4).

$\square$

**Lemma D.8.**
$$\lambda_{\min}(\widehat{\Theta}) \geq \lambda_{\min}\left(\mathcal{X}\mathcal{X}^\top\right)/4.$$

*Proof.* Notice that for ReLU activation $\phi$, a simple fact is that $\phi'(ax) = \phi'(x)$ holds for any $x \in \mathbb{R}$ given that $a > 0$. Therefore,

$$\begin{aligned}
\widehat{\Theta}_{ij} &= x_i^\top x_j \mathbb{E}_{w\sim\mathcal{N}(0,\frac{1}{d}\mathbb{I}_d)}\left[\phi'(w^\top x_i)\phi'(w^\top x_j)\right] \\
&= x_i^\top x_j \mathbb{E}_{w\sim\mathcal{N}(0,\mathbb{I}_d)}\left[\phi'(w^\top x_i)\phi'(w^\top x_j)\right] \\
&= \frac{x_i^\top x_j(\pi - \arccos(x_i^\top x_j))}{2\pi} \\
&= \frac{x_i^\top x_j}{4} + \frac{x_i^\top x_j}{2\pi}\arcsin(x_i^\top x_j) \\
&= \frac{x_i^\top x_j}{4} + \frac{1}{2\pi}\sum_{k=0}^{\infty}\frac{(2k)!}{4^k(k!)^2(2k+1)}(x_i^\top x_j)^{2k+2}.
\end{aligned}$$

Then

$$\widehat{\Theta} = \frac{\mathcal{X}\mathcal{X}^\top}{4} + \frac{1}{2\pi} \sum_{k=0}^{\infty} \frac{(2k)!}{4^k (k!)^2 (2k+1)} \left(\mathcal{X}\mathcal{X}^\top\right)^{\circ(2k+2)}$$

$$= \frac{\mathcal{X}\mathcal{X}^\top}{4} + \frac{1}{2\pi} \sum_{k=0}^{\infty} \frac{(2k)!}{4^k (k!)^2 (2k+1)} \left((\mathcal{X}^\top)^{\odot(2k+2)}\right)^\top (\mathcal{X}^\top)^{\odot(2k+2)},$$

where $\circ$ is the element-wise product, and $\odot$ is the Khatri-Rao product[6].

Since $\left((\mathcal{X}^\top)^{\odot(2k+2)}\right)^\top (\mathcal{X}^\top)^{\odot(2k+2)}$ is positive semidefinite, we have

$$\lambda_{\min}(\widehat{\Theta}) \geq \lambda_{\min}\left(\mathcal{X}\mathcal{X}^\top\right)/4,$$

which completes the proof.

$\square$

In the next lemma, we adopt an inequality from (Montgomery-Smith, 1990).

**Lemma D.9.** *If $\{\sigma_i\}_{i=1}^n$ are i.i.d. drawn from $U\{-1, 1\}$, then for any $x = (x_1, \ldots, x_n)^\top \in \mathbb{R}^n$, with probability at least $1 - \delta$ over $\sigma$,*

$$\left|\sum_{i=1}^n \sigma_i x_i\right| \leq \sqrt{2\log(2/\delta)} \, \|x\| \, .$$

The following lemma gives a sharp bound for a Chi-square variable, which is from (Laurent & Massart, 2000, Lemma 1).

**Lemma D.10.** *Let $(Y_1, \ldots, Y_D)$ be i.i.d. Gaussian variables, with mean 0 and variance 1. Then with probability at least $1 - \delta$ over $Y$,*

$$\sum_{i=1}^D Y_i^2 \leq D + 2\sqrt{D\log\left(\frac{1}{\delta}\right)} + 2\log\left(\frac{1}{\delta}\right).$$

The next lemma is quoted from (Arora et al., 2019, Lemma 5.4), giving an upper bound for the empirical Rademacher complexity if one has an accurate estimate for the distance with respect to each hidden unit.

**Lemma D.11.** *Given $R > 0$, consider the following function class*

$$\mathcal{F} = \left\{ x \mapsto f(w, x) : \left\|W_{1,r} - W_{1,r}^{(0)}\right\| \leq R(\forall r \in [m]), \left\|W_1 - W_1^{(0)}\right\|_F \leq B \right\}$$

*with $W_{1,r} \in \mathbb{R}^d$ the $r$-th row of $W_1$. Then for an i.i.d. sample $S = \{x_1, \ldots, x_n\}$ and every $B > 0$, with probability at least $1 - \delta$ over the random initialization, the empirical Rademacher complexity is bounded as:*

$$\mathcal{R}_S(\mathcal{F}) \leq \frac{B}{\sqrt{2n}} \left(1 + \left(\frac{2\log(2/\delta)}{m}\right)^{1/4}\right) + 2R^2 \sqrt{md} + R\sqrt{2\log(2/\delta)}.$$

Now we are ready to prove Theorem 5.

---

[6]For $A = (a_1, \ldots, a_n) \in \mathbb{R}^{m \times n}, B = (b_1, \ldots, b_n) \in \mathbb{R}^{p \times n}$, then $A \odot B = [a_1 \otimes b_1, \ldots, a_n \otimes b_n]$, where $\otimes$ is the Kronecker product.

*Proof of Theorem 5.* For Step 1. By Lemma D.7, if $m \geq \text{poly}\left(n, \lambda_{\min}^{-1}(\widehat{\Theta}), \delta^{-1}\right)$, then with probability at least $1 - \delta$ over the random initialization,

$$
\begin{aligned}
\left\|\nabla \mathcal{L}_n(w^{(t)})\right\| &= \frac{1}{n}\left\|\sum_{i=1}^{n}\left(f(w, x_i) - y_i\right)\nabla f(w^{(t)}, x_i)\right\| \\
&= \frac{1}{n}\left\|\nabla f(w^{(t)}, \mathcal{X})^\top \left(f(w^{(t)}, \mathcal{X}) - \mathcal{Y}\right)\right\| \\
&= \frac{1}{n}\sqrt{\left(f(w^{(t)}, \mathcal{X}) - \mathcal{Y}\right)^\top \nabla f(w^{(t)}, \mathcal{X})\nabla f(w^{(t)}, \mathcal{X})^\top \left(f(w^{(t)}, \mathcal{X}) - \mathcal{Y}\right)} \\
&\geq \sqrt{\frac{2\lambda_{\min}(\Theta(t))}{n}}\sqrt{\mathcal{L}_n(w^{(t)})} \\
&\geq \sqrt{\frac{\lambda_{\min}(\widehat{\Theta})}{n}}\sqrt{\mathcal{L}_n(w^{(t)})}
\end{aligned}
$$

holds for any $t \geq 0$, which means that $\mathcal{L}_n(w^{(t)})$ satisfies the Uniform-LGI for any $t \geq 0$ with

$$
c_n = \sqrt{\lambda_{\min}(\widehat{\Theta})/n}, \quad \theta_n = 1/2.
$$

For Step 2. By equation (13), we can directly get $\mathcal{L}_n(w^{(t)})$ converges to zero with a linear convergence rate:

$$
\mathcal{L}_n(w^{(t)}) \leq \exp\left(-\lambda_{\min}(\widehat{\Theta})t/n\right)\mathcal{L}_n(w^{(0)}).
$$

For Step 3. By equation (12) and Lemma D.8, the length $\text{len}(w^{(0)}, \infty)$ can be bounded as

$$
\text{len}(w^{(0)}, \infty) \leq r_n(w^{(0)}) = 2\sqrt{\frac{n\mathcal{L}_n(w^{(0)})}{\lambda_{\min}(\widehat{\Theta})}} \leq 4\sqrt{\frac{n\mathcal{L}_n(w^{(0)})}{\lambda_{\min}(\mathcal{X}\mathcal{X}^\top)}}.
$$

By the property of the target function, we have

$$
\begin{aligned}
\sqrt{n\mathcal{L}_n(w(0))} &= \sqrt{\frac{1}{2}\sum_{i=1}^{n}\left(\frac{1}{\sqrt{m}}w_2^\top \phi(W_1^{(0)}x_i) - y_i\right)^2} \\
&\leq \sqrt{\sum_{i=1}^{n}\left(\frac{1}{\sqrt{m}}w_2^\top \phi(W_1^{(0)}x_i)\right)^2 + y_i^2} \\
&\leq \sqrt{\sum_{i=1}^{n}\left(\frac{1}{\sqrt{m}}w_2^\top \phi(W_1^{(0)}x_i)\right)^2} + \sqrt{\sum_{i=1}^{n}y_i^2} \\
&\leq \sqrt{\frac{1}{m}\sum_{i=1}^{n}\left(w_2^\top \phi(W_1^{(0)}x_i)\right)^2} + c^*\sqrt{\lambda_{\max}(\mathcal{X}\mathcal{X}^\top)}.
\end{aligned}
$$

Since the entries of $w_2$ are drawn i.i.d. from $U\{-1, 1\}$, then by Lemma D.9, for each $i \in [n]$, with probability at least $1 - \delta/6n$ over $w_2$,

$$
\left(w_2^\top \phi(W_1^{(0)}x_i)\right)^2 \leq 2\log\left(\frac{12n}{\delta}\right)\left\|\phi(W_1^{(0)}x_i)\right\|^2.
$$

Taking the union bound over all $i = 1, 2, \ldots, n$, we have with probability at least $1 - \delta$ over the random initialization,

$$
\begin{aligned}
\sqrt{\frac{1}{m} \sum_{i=1}^{n} \left( w_2^\top \phi(W_1^{(0)} x_i) \right)^2} &\leq \sqrt{\frac{2 \log(12n/\delta)}{m} \sum_{i=1}^{n} \left\| \phi(W_1^{(0)} x_i) \right\|^2} \\
&= \sqrt{\frac{2 \log(12n/\delta)}{m}} \left\| \phi \left( W_1^{(0)} \mathcal{X}^\top \right) \right\|_F \\
&\leq \sqrt{\frac{2 \log(12n/\delta)}{m}} \left\| W_1^{(0)} \mathcal{X}^\top \right\|_F \\
&\leq \sqrt{\frac{2 \log(12n/\delta)}{m}} \left\| W_1^{(0)} \right\|_F \| \mathcal{X} \| \\
&= \sqrt{\frac{2 \log(12n/\delta)}{m}} \left\| W_1^{(0)} \right\|_F \sqrt{\lambda_{\max}(\mathcal{X}\mathcal{X}^\top)}.
\end{aligned}
\tag{30}
$$

For the Gaussian random matrix $W_1^{(0)} \sim \mathcal{N}(0, \frac{1}{d}\mathbb{I}_{m \times d})$, by Lemma D.10, we have with probability at least $1 - \delta/6$ over the random initialization,

$$
\frac{\left\| W_1^{(0)} \right\|_F^2}{m} \leq 1 + 2\sqrt{\frac{\log(6/\delta)}{md}} + \frac{2 \log(6/\delta)}{md}.
\tag{31}
$$

Taking the union bound of equation (19), (20), (30) and (31), if $m \geq \mathrm{poly}\left(n, \lambda_{\min}^{-1}(\widehat{\Theta}), \delta^{-1}\right)$, then with probability at least $1 - \tau^{d-n+1} - \tau^d - 5\delta/6$ over the samples and random initialization $\mathcal{I}$,

$$
\begin{aligned}
\sup_{w^{(0)} \in \mathcal{I}, (x,y) \in \mathcal{D}} r_n(w^{(0)}) &\leq \sup_{w^{(0)} \in \mathcal{I}, (x,y) \in \mathcal{D}} 4\sqrt{\frac{n\mathcal{L}_n(w^{(0)})}{\lambda_{\min}(\mathcal{X}\mathcal{X}^\top)}} \\
&\leq \mathcal{O}\left( \sqrt{\left(1 + \sqrt{\frac{\log(1/\delta)}{md}} + \frac{\log(1/\delta)}{md}\right) \log(n/\delta)} \frac{\sqrt{n} + \sqrt{d} + \sqrt{\log(1/\delta)}}{\sqrt{d} - \sqrt{n-1}} \right) \\
&\leq \mathcal{O}\left( \sqrt{\log(n/\delta)} \frac{\sqrt{\gamma_1} + 1 + \sqrt{\log(1/\delta)/n}}{1 - \sqrt{\gamma_1}} \right) \\
&\leq \mathcal{O}\left( \sqrt{\log(n/\delta)} \right).
\end{aligned}
$$

Therefore, with probability at least $1 - \tau^{d-n+1} - \tau^d - 5\delta/6$ over the samples and random initialization,

$$
\frac{r_n(w^{(0)})}{\sqrt{n}} \leq \mathcal{O}\left( \sqrt{\frac{\log(n/\delta)}{n}} \right).
\tag{32}
$$

For the $r$-th row of $W_1^{(t)}$, we begin to bound the distance $\left\| W_{1,r}^{(t)} - W_{1,r}^{(0)} \right\|$ for each $r \in [m]$.

Notice that

$$\left\| \frac{dW_{1,r}^{(t)}}{dt} \right\| = \left\| \nabla_{W_{1,r}^{(t)}} \mathcal{L}_n(w^{(t)}) \right\|$$

$$= \left\| \frac{1}{n} \sum_{i=1}^{n} \left( f(w^{(t)}, x_i) - y_i \right) \frac{1}{\sqrt{m}} w_{2,r} \phi'(W_{1,r}^{(t)} x_i) x_i \right\|$$

$$\leq \frac{1}{n\sqrt{m}} \sum_{i=1}^{n} \left| f(w^{(t)}, x_i) - y_i \right|$$

$$\leq \frac{1}{\sqrt{nm}} \left\| f(w^{(t)}, \mathcal{X}) - \mathcal{Y} \right\|$$

$$\leq \sqrt{\frac{2\mathcal{L}_n(w^{(0)})}{m}} \exp\left( -\lambda_{\min}(\widehat{\Theta}) t / 2n \right).$$

Hence,

$$\left\| W_{1,r}^{(t)} - W_{1,r}^{(0)} \right\| \leq \int_0^t \left\| \frac{dW_{1,r}^{(s)}}{ds} \right\| ds \leq \frac{2n}{\lambda_{\min}(\widehat{\Theta})} \sqrt{\frac{2\mathcal{L}_n(w^{(0)})}{m}} = \sqrt{\frac{2n}{m\lambda_{\min}(\widehat{\Theta})}} r_n(w^{(0)}).$$

Now we apply Lemma D.11 with $B = r_n(w^{(0)}), R = \sup_{w^{(0)} \in \mathcal{I}, (x,y) \in \mathcal{D}} \sqrt{\frac{2n}{m\lambda_{\min}(\widehat{\Theta})}} r_n(w^{(0)})$, we get with probability at least $1 - \delta/12$ over the random initalization,

$$\mathcal{R}_S(\mathcal{F}) \leq \frac{B}{\sqrt{2n}} \left( 1 + \left( \frac{2\log(24/\delta)}{m} \right)^{1/4} \right) + 2R^2\sqrt{md} + R\sqrt{2\log(24/\delta)}.$$

Then by equation (32), if $m \geq \text{poly}\left( n, \lambda_{\min}^{-1}(\widehat{\Theta}), \delta^{-1} \right)$, then with probability at least $1 - \tau^{d-n+1} - \tau^d - 11\delta/12$ over the samples and random initialization,

$$\mathcal{R}_S(\mathcal{F}) \leq \mathcal{O}\left( \sqrt{\frac{\log(n/\delta)}{n}} \right).$$

Finally, by Lemma C.1, we have with probability at least $1 - \tau^{d-n+1} - \tau^d - \delta$ over the samples and random initialization,

$$\mathbb{E}_{(x,y)\sim\mathcal{D}}\left[ \tilde{\ell}\left( f(w^{(\infty)}, x), y \right) \right] \leq \mathcal{O}\left( \sqrt{\frac{\log(n/\delta)}{n}} \right),$$

for some constant $\tau \in (0,1)$ that depend only on the subgaussian moment of the entries.

This completes the proof.

$\square$

