# OpenReview forum: "Short optimization paths lead to good generalization"
_ICLR.cc/2022/Conference — ICLR 2022 Submitted_

### Official Review · Reviewer_shcV · 2021-11-03

**Correctness:** 3
**Technical Novelty And Significance:** 3
**Empirical Novelty And Significance:** Not applicable
**Recommendation:** 5
**Confidence:** 4

**Main Review:**

The paper is well organized and easy to follow. My main concern is regarding the claim "short optimization paths lead to good generalization" which I feel misleading. The reason is that

a. The upper bound $r_n(w)$ is not path-length related. In contrast, it only depends on the function value gap $L_n(w) - L_n^*$. In other words, we first upper bound the path length by the function value gap, which is expected under Łojasiewicz gradient inequality, then use the later function value gap to derive the generalization bound. Unless the path-length is used in the proof of Theorem 2, I wouldn't call it a length-based generalization bound. Please provide more details on this point.

b. I am not sure whether the following case is possible: there are $w_0$ and $w_1$ that for any $L_n$, the gradient flow from $w_0$ always pass through $w_1$. In this case, they always share the same solution but the path length of $w_1$ is always shorter than the path length of $w_0$. Even though this might be impossible, what I want to say is I don't find the shorter length-path as the cause (in terms of causality) of better generalization, instead, I think the function value gap is the real cause of it, which I believe is well studied in the literature. (it might be wrong but I am happy to further discuss it)

Overall, I can't see how explicitly the length of optimization path influence the generalization bound in the current analysis, instead it is based on the function value gap. I am willing to raise my score if my concern is addressed in the rebuttal.

**Summary Of The Paper:**

The paper provides a novel generalization bound for the gradient flow equation related to the length of the optimization path. The bound is valid for loss function that locally satisfies Łojasiewicz gradient inequality, which is applicable to different machine learning models such as underdetermined $\ell_p$ linear regression, kernel regression, and overparameterized two-layer ReLU neural networks. Explicit derivations are provided for these three models to show that the length-based generalization bound is non-vacuous.

**Summary Of The Review:**

My major concern is how explicitly the length of optimization path influence the generalization bound in the current analysis, instead it is based on the function value gap.

---

> ### Author Response · Authors · 2021-11-18
> **Authors' response**
>
> 1. ''The upper bound $r_n(w)$ is not path-length related. In contrast, it only depends on the function value gap.
> In other words, we first upper bound the path length by the function value gap, which is expected under Łojasiewicz gradient inequality, then use the later function value gap to derive the generalization bound. Unless the path-length is used in the proof of Theorem 2, I wouldn't call it a length-based generalization bound. Please provide more details on this point.''
>
>    * We would like to point out that Theorem 2 is precisely a length-based generalization bound.
>     The intuition in Theorem 2 is that, if the initialization $w^{(0)}$ is random/independent of the training data, then it has a minimal generalization gap.
>     If the trajectory length is small, then $w^{(\infty)}$ is necessarily close to $w^{(0)}$, thus having small generalization gap as well.
>     In fact, Theorem 2 gives a length-based generalization bound once we have an explicit length estimate.
>     The role of Theorem 1 is to provide a sufficient condition to obtain such an estimate,
>     by bounding the path length in terms of the function value gap.
>     Together, we connect optimization and generalization.
>
>      **We have improved the precision of Theorem 2 and added Remark 1 to explain the main idea above.**
>
>
> 2. ''I am not sure whether the following case is possible: there are $w_0$
> and $w_1$ that for any $L_n$, the gradient flow from $w_0$
> always pass through $w_1$. In this case, they always share the same solution but the path length of $w_1$ is always shorter than the path length of $w_0$. Even though this might be impossible, what I want to say is I don't find the shorter length-path as the cause (in terms of causality) of better generalization, instead, I think the function value gap is the real cause of it, which I believe is well studied in the literature.''
>
>    * First, we emphasize that the claim that the generalization of $w_1$ is better than the generalization of $w_0$ cannot be deduced from Theorem 2.
>     Note that the generalization bound in Theorem 2 is a high probability length-based bound over the initialization and training data, and we require the initialization to be independent of the random choice of the training data.
>     The situation where one realization of this random initialization lies in the gradient descent path of another realization is in general a 0-probability event, hence our bounds do not cover these cases.
>     For this reason, all of our numerical evaluations focus on the statistics of trajectory lengths: in each case, we vary some initialization/model configurations so that the optimization trajectories vary in a statistically significant way, from which we observed that the corresponding generalization behavior changes in the same manner.
>     See Figure 1 and our additional numerical results in Appendix A.1 \& A.2.
>
>      **We have revised the paper to address this in the discussion of Theorem 2.**
>
>    * Second, as we mentioned before, Theorem 2 itself is a length-based generalization bound, and Theorem 1 serves as a sufficient condition to provide an explicit length estimate in terms of the function value gap. As long as one has an explicit length estimate for the trajectory length, one can obtain a generalization estimate by replacing $R_{n, \delta}$ with the new length estimate.
>
>      **In light of your concerns, we have modified the statement of Theorem 2 and added explanations for the roles of Theorem 1 \& 2 before and after Theorem 2.**

---

> > ### Comment · Reviewer_shcV · 2021-11-23
> > **Re: Authors' response**
> >
> > I thank the provided clarification and reformulation of Theorem 2. After the revision, I agree that that Theorem 2 is a length-based generalization bound. Several questions remain:
> > 1. Now the question becomes in which situation we can obtain useful the length-based generalization bound. The Theorem 1 is one use case where the loss function is Uniform-LGI. However this is not that interesting as we can directly use the function value gap to bound the generalization error. A less straightforward example would be necessary to demonstrate the strength of the results
> > 2. Imagine we want to apply this generalization bound, then we need to bound the length bound $R_{n, \delta}$. Except some case such as Uniform-LGI, where the length can be bounded by the function value gap, the constant $R_{n, \delta}$ is not explicitly computable. Furthermore, it is not clear that $R_{n, \delta}$ is independent of $n$.
> > Overall, I believe further clarification is required.

---

> > > ### Author Response · Authors · 2021-11-23
> > > **Reply to Reviewer shcV**
> > >
> > > We thank the reviewer for the comments. The main concern is on the application of Theorem 2. We attempt to clarify this below:
> > >
> > >   1. ''Now the question becomes in which situation we can obtain useful the length-based generalization bound. The Theorem 1 is one use case where the loss function is Uniform-LGI. However this is not that interesting as we can directly use the function value gap to bound the generalization error. A less straightforward example would be necessary to demonstrate the strength of the results''
> > >
> > >    * First, we would like to point out that once the loss function satisfies the Uniform-LGI, one can obtain a length estimate by Theorem 1.    We have shown the applications on three machine learning models: $\ell_p$ linear regression (Theorem 3), $\ell_p$ kernel regression (Theorem 4) and overparameterized two-layer neural networks (Theorem 5).   These bounds match or expand the previous results.
> > >
> > >
> > >   * Second, we emphasize that the function value gap is on the *training loss*, but *not* on the expected loss. Therefore, it does not in general translate to a bound on generalization.
> > >     Moreover,
> > >     under the Uniform-LGI condition, the estimation of the path length depends not only on the function value gap, but also the Uniform-LGI cosntants $c_n$ and $\theta_n$. The asymptotic analysis of $c_n$ and $\theta_n$ is also interesting and non-trivial. Therefore, we think our three applications demonstrate the strength of our framework.
> > >
> > > 2. ''Imagine we want to apply this generalization bound, then we need to bound the length bound $R_{n, \delta}$. Except some case such as Uniform-LGI, where the length can be bounded by the function value gap, the constant $R_{n, \delta}$ is not explicitly computable. Furthermore, it is not clear that $R_{n, \delta}$ is independent of $n$. Overall, I believe further clarification is required.''
> > >
> > >   * First, we emphasize that the main contribution of this paper is to propose a framework that connects optimization and generalization.
> > >     The key component is the Uniform-LGI property.
> > >     By combining Theorem 1 and Theorem 2, we have shown three applications to obtain generalization estimates on $\ell_p$ linear/kernel regression and overparameterized two-layer neural networks.
> > >     These results match or expand the type of scenarios where we can rigorously establish the phenomenon of benign overfitting.
> > >     Path length estimates for other types of loss function property (other than Uniform-LGI) is not the core scope of the current paper.
> > >
> > >    * Second, as we mentioned, $R_{n, \delta}$ depends not only on the training loss value gap, but also the Uniform-LGI constants $c_n$ and $\theta_n$.
> > >     The asymptotic analysis of $c_n$ and $\theta_n$ is problem-dependent, and we have given explicit computations for $R_{n, \delta}$ with respect to different applications. See equation (21) for $\ell_p$ linear regression, equation (24) for RBF kernel regression, equation (29) for inner product kernel regression, and equation (32) for overparemtereized two-layer neural networks.

---

### Official Review · Reviewer_xkEt · 2021-11-03

**Correctness:** 4
**Technical Novelty And Significance:** 3
**Empirical Novelty And Significance:** Not applicable
**Recommendation:** 6
**Confidence:** 3

**Main Review:**

The paper inscribes in the relevant research direction aiming at understanding the interplay between efficient optimization and good generalization performances by identifying complexity measures that are implicitly minimised during training.

The analysis is sound and presented in a clear way. Overall, I found the results interesting and worth of publication although some improvements are needed.

In my opinion, the main weakness of the work regards the numerical experiments. I find that the authors could have provided better evidence than Figure 1. It is not clear to me whether each point in Figure 1 comes from an average (at fixed initialisation variance) or is the result of a single simulation starting from random Gaussian initialization with that variance. The first option does not make much sense to me, so I am assuming the second. However, in this case, since the initialization is random I do not understand why the authors need to change also the variance instead of comparing the optimization paths starting from different initializations from the same distribution. The way the figure is presented suggests that the length of the optimization path trivially depends on initialization, and similarly the generalization gap.

Moreover, all the experiments are performed for a learning rate $\eta=0.05$. Since the theory is valid in the limit of gradient flow, I would be interested in seeing how Figure 1 changes for different learning rates.

Finally, it would be interesting to test the generality of the proposed framework by checking the relation between short optimization paths and good generalization in more realistic data model. Has this relation been previously observed in applications?

**Summary Of The Paper:**

The authors study the connection between optimization and generalization for gradient flow (GF) on loss functions that satisfy a global version of the Lojasiewicz gradient inequality. Under this assumption, they prove convergence of GF to a global minimum and they find an upper bound on the optimization length — measured as the integral of the $\ell_2-$norm of the gradient from time $t=0$ to the final time. This upper bound depends on the specific choice of the loss function and on the number of samples. With an additional assumption on the hypothesis class (encompassing, e.g., linear shallow networks, two-layer networks) the first result is used to derive an upper bound on the generalization gap. The bound depends on the choice of the loss function, its initial value, and the length of the optimisation path. This leads to the main result that shorter optimization paths induce smaller generalization gap. The authors apply this result to three models (underdetermined $\ell_p$ linear regression, kernel regression, and overparametrized two-layer networks with ReLU activation) with a given target function. They compute non-asymptotic expressions for the generalization bounds at fixed ratio between sample size and ambient dimension, and show that in these cases the bounds are non-vacuous when the dimension increases.


**Summary Of The Review:**

I find that this paper is worth of publication since it provides an interesting contribution to a relevant research direction in the theory of machine learning. However, I believe that the authors must improve their numerical results to provide a convincing final version of the work.

---

> ### Author Response · Authors · 2021-11-18
> **Authors' response**
>
> 1. ''Since the initialization is random I do not understand why the authors need to change also the variance instead of comparing the optimization paths starting from different initializations from the same distribution. The way the figure is presented suggests that the length of the optimization path trivially depends on initialization, and similarly the generalization gap.''
>
>     * First, we point out that our generalization bound in Theorem 2 is a high probability bound over the initialization and the training set.
>     Each set value of $\sigma$ corresponds to one such initialization scheme.
>     Intuitively, to demonstrate our high probability result,
>     it is not meaningful to compare a small number of specific trajectory realizations.
>     Rather, we need to compare the statistics of these trajectories.
>     If we use the same $\sigma$, then the path lengths will not vary significantly in the statistical sense.
>     Therefore, we use different $\sigma$ (and also now, different means) to induce optimization path length variations that are statistically significant.
>     These can then support our high probability results.
>
>       **To make this point clearer, we have modified the statement of Theorem 2, where we use $R_{n, \delta}$ to represent the uniform length estimate depending on the initialization scheme.**
>
>    * Second, we would like to clarify that the length of the optimization path is a sufficient but not necessary condition to guarantee the generalization.
>     Figure 1 shows that the path length can be a good indicator of generalization ability.
>
>       **To better illustrate the relation between optimization path length and generalization, we have provided additional numerical experiments for more initialization schemes (Appendix A.1).**
>
> 2. ''Moreover, all the experiments are performed for a learning rate $\eta = 0.05$. Since the theory is valid in the limit of gradient flow, I would be interested in seeing how Figure 1 changes for different learning rates.''
>
>     **In light of your comments, we have added additional experiments for different learning rates in Appendix A.1.** The results support our result that short optimization paths lead to good generalization for different learning rate.
>
> 3. ''It would be interesting to test the generality of the proposed framework by checking the relation between short optimization paths and good generalization in more realistic data model. Has this relation been previously observed in applications?''
>
>     **We have provided numerical evaluations on MNIST dataset in Appendix A.2,**  the results suggest the optimization path bridges the connection between optimization and generalization in a more realistic data model as well. To the best of our knowledge, this relation has not been previously observed.

---

> > ### Comment · Reviewer_xkEt · 2021-11-30
> > **Response to authors**
> >
> > I thank the authors for addressing my concerns providing useful clarifications. I have appreciated the effort in improving the manuscript, that I recommend for publication. Therefore, I am keeping my score to 6.

---

### Official Review · Reviewer_8Scj · 2021-11-03

**Correctness:** 4
**Technical Novelty And Significance:** 2
**Empirical Novelty And Significance:** Not applicable
**Recommendation:** 5
**Confidence:** 4

**Main Review:**

This paper is well organized following the approach from optimization to generalization. The theorems and proofs are clearly stated and easy to follow. However, my main concern is that it seems to put two separate results (one for optimization and the other for generalization) together directly, and the novelty of each of the two results may not be significant enough. Also, the results may not support the claim “short optimization paths lead to good generalization” exactly.

For the optimization part, the definition of the Uniform-LGI (with parameter $\theta$) is inspired by the original Lojasiewicz inequality, and the proof of convergence is straightforward and similar to Bolte et al. (2007). Nevertheless, I think it is good to introduce the result to the community as an extension to the widely-used PL condition. The sublinear convergence rate is also interesting. It would be better to have more applications satisfying the Uniform-LGI but not the PL condition. In Section 4 only the $l_p$ linear regression has $\theta > 1/2$, and the other two models still fall into the PL condition.

For the generalization part, the estimation of generalization error follows the Rademacher approach. The proof is nontrivial while the idea is not complicated: the Rademacher complexity scales with the diameter of the parameter set, i.e., the optimization path length. To claim “short optimization paths lead to good generalization”, I may expect a path-based generalization bound; ~~I think the good generalization here comes more from the good “local landscape” (Uniform-LGI) rather than the “path”.~~ (Edit: my statement was not accurate—Theorem 2 does not require the Uniform-LGI. I mean the generalization bound does not depend on the whole optimization path, but only the endpoint, and the generalization is good if the endpoint falls in a small region.)

In addition, I think the numerical result in Figure 1 may not illustrate the relationship between the optimization path length and generalization—the larger $\sigma$ in the initialization (Appendix A) leads to both larger optimization path length and larger generalization gap. I think it would be more fair to compare them with the same $\sigma$ in the initialization.

**Update**

I appreciate the detailed response from the authors and the careful improvement of the manuscript. Here is my summary of the contributions of the paper:
* Optimization: it is good to propose the Uniform-LGI as an extension of the PL condition and show the sublinear rate. However, the Uniform-LGI is similar to the original Lojasiewicz inequality, and the proof is straightforward (similar to Bolte et al., 2007?).
* Generalization: Theorem 2 is equivalent to the following: given a region $S$ with radius $R$, with probability $1 - \delta$ over the training samples, for all $w \in S$, the generalization gap $$L_D(w) - L_n(w) \le \frac{A R + B \sqrt{3(p + q) + \log(2 / \delta)}}{\sqrt n}$$ where $A$ depends on the Lipschitz constants in $S$ and $B$ depends on the upper bound of $l$. Similar approach is taken by some previous work (e.g., Allen-Zhu et al., 2019); nevertheless, it is good to state the result precisely. I think the result would be better summarized as “small parameter region leads to good generalization bound”, and this is not surprising as the Rademacher complexity increases w.r.t. $R$.

Therefore, I would like to keep my original score given my concern about the significance of the novelty.

**Summary Of The Paper:**

This paper proposes a framework to analyze both optimization and generalization properties under the Uniform-LGI condition (Def 1). From my understanding, the main results consist of two parts:

* Optimization: define the Uniform-LGI as an extension of the PL condition, prove the corresponding convergence result with a sublinear rate, and bound the optimization path length.
* Generalization: use the Rademacher complexity to estimate the generalization error. The Rademacher complexity scales with the diameter of the parameter set, thus can be bounded by the optimization path length, which connects with the optimization results.

Then the paper apply this framework to three application models: first establish the Uniform-LGI, then calculate the optimization path length and estimate the generalization error.

**Summary Of The Review:**

The paper is well organized and clearly written. It is interesting to introduce the Uniform-LGI as an extension to the PL condition. However, I think the novelty of each of the two main results is not significant enough, and putting them together may not support the claim “short optimization paths lead to good generalization” exactly.

---

> ### Author Response · Authors · 2021-11-18
> **Authors' response**
>
> 1. ''It seems to put two separate results (one for optimization and the other for generalization) together directly, and the novelty of each of the two results may not be significant enough.''
>
>     * Novelty of Theorem 1:
>     Theorem 1 shows that under the uniform LGI condition,
>     the optimization path length can be bounded by the initial loss function value
>     and the uniform LGI constants.
>     Previous work [1] showed similar results under the more restrictive PL condition.
>     Here, we show that the uniform LGI condition is sufficient to guarantee length estimates,
>     and the key difference from those using the PL condition is that the convergence rate is
>     not necessarily linear.
>     As far as we are aware, this is a new result concerning optimization path length estimation
>     and convergence rate analysis.
>
>      * Novelty of Theorem 2:
>     Theorem 2 is a path length based generalization bound.
>     This differs from the work of [2] in that our bound holds for any hypothesis $f(w, \cdot)$ that satisfies the equation (3), but not only for two-layer ReLU neural networks.
>
>     * Most importantly, the key novelty of this paper is the combined results of Theorems 1 \& 2.
>     Previous generalization bounds that connects with optimization are based on minimum norm, or max margin solutions [3, 4].
>     These trained configurations require very special structures of the loss function/data to achieve.
>     In contrast, the generalization bound in Theorem 2 only relies on a path length estimate,
>     and Theorem 1 gives a sufficient condition to ensure that such an estimate exists.
>     Thus, the result here does not require specific setups, and can be applied to analyze a variety of model in exactly the same workflow, as we demonstrated in the practical applications.
>     We believe that this is a central novelty of this paper.
>
>       __We have added some discussion of this in Remark 1 to emphasize this point.__
>
> 2. ''Also, the results may not support the claim “short optimization paths lead to good generalization” exactly.''
>
>    * Theorem 2 gives a length-based generalization bound $\mathcal{O} (R / \sqrt{n})$, which indicates that when the length estimate is small, for example, with order $\mathcal{O}(1)$, then we can get a non-vacuous bound $\mathcal{O}(1 / \sqrt{n})$, which is a good generalization bound.
>     Thus, the precise statement of Theorem 2 is that short optimization path leads to a good generalization *bound*.
>
>       __We have made this point clearer in the discussion following Theorem 2.__
>
> 3. ''It would be better to have more applications satisfying the Uniform-LGI but not the PL condition. In Section 4 only the
>  linear regression has $\theta > 1 / 2$, and the other two models still fall into the PL condition.''
>
>    * In view of the reviewer's comments, __we have added the analysis for kernel regression with $\ell_p$ loss in Theorem 4__. For $p > 2$, the model satisfies the Uniform-LGI condition but not PL condition.
>
> 4. ''To claim “short optimization paths lead to good generalization”, I may expect a path-based generalization bound; I think the good generalization here comes more from the good “local landscape” (Uniform-LGI) rather than the “path”.''
>
>     * First, Theorem 2 is precisely a path-based generalization bound.
>     It shows that if the optimization trajectory is short ($R$ is small), then this leads to small generalization error.
>     We emphasize that Theorem 2 does not assume any landscape properties.
>       The Uniform-LGI condition concerning the landscape only enters Theorem 1, which gives a *sufficient*
>     but *not necessary* condition to ensure short optimization paths.
>       In essence, while uniform-LGI condition (a landscape property) is used to derive the generalization estimates,
>     it enters only through the estimation of path lengths.
>     Thus, the statement “short optimization paths lead to good generalization” is supported by our main results.
>
> 5. ''I think the numerical result in Figure 1 may not illustrate the relationship between the optimization path length and generalization—the larger $\sigma$ in the initialization (Appendix A) leads to both larger optimization path length and larger generalization gap. I think it would be more fair to compare them with the same $\sigma$ in the initialization.''
>
>     * First, we emphasize that $\sigma$ is not the only factor that determines the trajectory length and the generalization error.
>     The mean of the initialization also matters.
>     __To clarify this statement, we have added more numerical results (the new Figure 1) by varying the mean and variance simultaneously.__
>     The experiment results suggest that short optimization paths is associated with good generalization,
>     and they are not simply affected by $\sigma$ alone.
>     This motivate us to consider length as a complexity to characterize the generalization.

---

> > ### Author Response · Authors · 2021-11-18
> > **Authors' response (continued)**
> >
> > * continued:
> >
> >    *  Second, the reason for presenting Figure 1 with different $\sigma$ is that, our generalization bound in Theorem 2 is a high probability bound over the initialization and the training set.
> >     Each set value of $\sigma$ corresponds to one such initialization scheme.
> >     Intuitively, to demonstrate our high probability result,
> >     it is not meaningful to compare a small number of specific trajectory realizations.
> >     Rather, we need to compare the statistics of these trajectories.
> >     If we use the same $\sigma$, then the path lengths will not vary significantly in the statistical sense.
> >     Therefore, we use different $\sigma$ (and also now, different means) to induce optimization path length variations that are statistically significant.
> >     These can then support our high probability results.
> >
> >        **To make this point clearer, we have modified the statement of Theorem 2, where we use $R_{n, \delta}$ to represent the uniform length estimate depending on the initialization scheme.**
> >
> >
> > References:
> >
> > [1]  Hamed Karimi, Julie Nutini, and Mark Schmidt. Linear convergence of gradient and proximal-gradientmethods under the polyak- lojasiewicz condition. In *Joint European Conference on Machine Learning and Knowledge Discovery in Databases*, pages 795–811. Springer, 2016.
> >
> > [2] Behnam Neyshabur, Zhiyuan Li, Srinadh Bhojanapalli, Yann LeCun, and Nathan Srebro. The role of over-parametrization in generalization of neural networks. *In International Conference on Learning Representations*, 2019.
> >
> > [3] Peter L Bartlett, Philip M Long, G ́abor Lugosi, and Alexander Tsigler.  Benign overfitting in linear regression. *Proceedings of the National Academy of Sciences*, 117(48):30063–30070, 2020.
> >
> > [4] Kaifeng Lyu and Jian Li.  Gradient descent maximizes the margin of homogeneous neural networks. In *International Conference on Learning Representations*, 2020

---

### Official Review · Reviewer_ZB5L · 2021-11-06

**Correctness:** 3
**Technical Novelty And Significance:** 3
**Empirical Novelty And Significance:** Not applicable
**Recommendation:** 6
**Confidence:** 3

**Main Review:**

The paper develops a convergence and generalization analysis of machine learning models under Uniform-LGI loss functions. The concept of Uniform-LGI loss functions extends the well-known PL condition in the optimization literature and as the paper shows leads to a nice theoretical setting for analyzing the optimization and generalization aspects of different machine learning models. The paper shows applications of the framework to norm-p regression and kernel regression problems as well as one-hidden-layer neural network learning problems.

Overall I think this is an interesting work which shows some insightful generalization results under Uniform-LGI loss functions. My main concern with the paper is that while the paper's title, abstract, and introduction claim that the analysis helps to understand the connection between the length of optimization and the generalization error of learned models, the theoretical results in Theorems 2-5 seem to only bound the generalization error of the final model and do not have anything to say about how the generalization error changes during training the model. I, therefore, suggest the authors either clearly explain how the theorems analyze the role of optimization length in the final generalization performance or change the title and introduction appropriately to reflect the main contributions of the work.

In addition to the above comment, I am wondering whether the gradient flow simplification in Equation (1) results in a realistic analysis of gradient-based optimization in non-convex deep learning problems. It seems to me that the analysis may not be suitable for stochastic gradient methods which have been shown to achieve a significantly better generalization performance than full-batch gradient descent modeled in the paper. As my final comment, the paper has no numerical results validating the theoretical generalization bounds over standard supervised learning problems. Therefore, it is unclear whether the theory results are useful to bound the generalization error of practical deep learning experiments.

**Summary Of The Paper:**

The paper studies generalization and optimization of kernel-based and one-hidden layer neural network models. The convergence guarantee is shown for a Uniform-LGI loss function in Theorem 1. Then, in Theorem 2 the paper proves a generalization guarantee assuming a particular parametric model in Equation (3). Finally, the paper applies these results to p-norm regression, kernel regression, and one hidden layer neural network learning problems.

**Summary Of The Review:**

The paper shows several interesting generalization results for Uniform-LGI loss functions. While the paper proves several insightful generalization bounds, it seems Theorems 2-5 do not explain the connection between the optimization length and generalization error. Also, the paper has no numerical results validating the generalization results.

---

> ### Author Response · Authors · 2021-11-18
> **Authors' response**
>
> 1. ''Explain how the theorems analyze the role of optimization length in the final generalization       performance.''
>
>    * The intuition for our results is as follows.
>     We show in theorem 2 that if the optimization trajectory length $\textsf{len}(w^{(0)}, \infty)$ is small,
>     then this implies a good generalization bound.
>     One can understand this by treating the trajectory length as a measure of complexity:
>     the initial point $w^{(0)}$ is random/independent of the training data, so it has a minimal generalization gap.
>     If the trajectory length is small, then $w^{(\infty)}$ is necessarily close to $w^{(0)}$, thus having small
>     generalization gap as well.
>     Theorem 1 gives, based on the uniform-LGI assumption, a sufficient condition to ensure that the trajectory
>     length can be estimated.
>     Hence, one can deduce a good generalization bound from Theorem 2.
>     Together, they connect optimization with generalization.
>
>      __We have added the above explanations before the statement of Theorem 2.__
>
> 2. '' Theorems 2-5 seem to only bound the generalization error of the final model and do not have anything to say about how the generalization error changes during training the model.''
>
>    * First, we emphasize that in this paper, by length we mean the trajectory length (path length) of the parameter evolution during training.
>     This is not the ``length of time'' used for training, as is usually analyzed in early-stopping type of algorithms.
>     Thus, our generalization bound concerns the weights of model trained to completion by empirical risk minimization ($w^{(\infty)}$).
>     The generalization of this final model is more important because it will be directly used on test data.
>
>    * Nevertheless, our framework can be extended to derive generalization estimates that evolves according to the length of time (number of epochs) of training. The approach is to give a generalization bound for *early stopping* when the loss value first reaches $\varepsilon \geq 0$.
>     The idea is that, when there exists $T > 0$ such that $\mathcal{L}_n(w^{T}) = \varepsilon$, then by the inequality (11) in the proof of Lemma B.1, we can get an upper bound for the length $\textsf{len} (w^{(0)}, T)$ in terms of $\mathcal{L}_n(w^{(0)}), \min_w \mathcal{L} (w), \varepsilon, c_n, \theta_n$. Finally, we can also get generalization bounds byTheorem 2 with our new length estimate.
>
>      __We have added additional theoretical results based on the above tow approaches in Appendix C.1.__
>
>
> 3. ''I am wondering whether the gradient flow simplification in Equation (1) results in a realistic analysis of gradient-based optimization in non-convex deep learning problems. It seems to me that the analysis may not be suitable for stochastic gradient methods.''
>
>    * First, when the step size (learning rate) of gradient is small, then the dynamics is close to the gradient flow.
>     This is guaranteed by the consistency of finite difference approximation of differential equations.
>     Thus, it is an idealized but relevant analytical setting.
>
>     * Second, continuous gradient flow allows one to employ more mathematical tools,
>     such as ODE solution methods, Gronwall inequalities, etc,
>     to obtain precise analytical results.
>
>      * Third, we emphasize that the main idea in this paper is not specific to deterministic gradient flow.
>     The key insight is our paper is to bridge a connection between optimization and generalization through path length estimates.
>     More precisely, Theorem 2 does not rely on deterministic gradient flow.
>     As long as one can obtain path length estimates (say from stochastic analysis of SGD), one can still
>     apply theorem 2 to obtain generalization estimates. Path length estimates for other types of training algorithms is an interesting future direction,
>     but not the core scope of the current paper.
>
>        __We have added the above discussion in Remark 1.__
>
> 4. ''The paper has no numerical results validating the theoretical generalization bounds over standard supervised learning problems. Therefore, it is unclear whether the theory results are useful to bound the generalization error of practical deep learning experiments.''
>
>    * __In light of your comments, we have added numerical evaluations for the generalization bounds on MNIST dataset in Appendix A.2.__
> The numerical results are consistent with our theoretical results that short optimization paths lead to good generalization.

---

> > ### Comment · Reviewer_ZB5L · 2021-11-30
> > **Thank you for the responses**
> >
> > I thank the authors for their thoughtful responses to my comments. While I still think the connection between generalization and optimization length can be analyzed more thoroughly, I think the overall contribution is sufficient for the paper's publication. I, therefore, keep my original weak accept rating.

---

### Author Response · Authors · 2021-11-18
**To all reviewers.**

*  We appreciate all the reviewers for their valuable suggestions and comments on our paper.

*  In the revised paper, we highlighted the main changes in blue, and in the following's response, we emphasize the changes in boldface.

*  The main concerns about the paper are two-fold.

   First, the connection between optimization and generalization is not clearly presented by our results.
    We emphasize that Theorem 2 is precisely a path-based generalization bound.
    It shows that as long as one has an explicit length estimate for the trajectory length, one can obtain a generalization estimate.
    That is to say, the role of Theorem 1 is to provide a sufficient but not necessary condition to ensure such a length estimate exists.
    In that sense, the statement ''short optimization paths lead to good generalization'' is supported by our results.


    The second main concern is that Figure 1 may not illustrate the relationship between the optimization path length and generalization.
    We would like to point out that the length estimate we consider in Theorem 2 is a uniform estimate for a specific initialization *method*, but not a point wise estimate that depends on a specific initialization *point*.
    Thus, the generalization performance for different initialization methods in Figure 1 motivates us to use path length to build its connection to optimization.

    **In this revision, we have modified the statement of Theorem 2, and added explanations for the roles of Theorem 1 \& 2 in the main text.**


* Based on each reviewer's concerns, we summarize the main changes of our paper below:

   + According to Reviewer ZB5L's comments, we have
     + **added more explanations of the intuition in the introduction;**
     + **provided additional theoretical results to bound the generalization error during training the model (Appendix C.1);**
     + **added numerical results validating our generalization bounds on MNIST dataset (Appendix A.2).**

  + According to Reviewer 8Scj's comments, we have
    + **modified the statement of Theorem 2, and added explanations for the roles and novelty of Theorem 1 \& 2 immediately after Theorem 2;**
    + **added the analysis for kernel regression with $\ell_p$ loss (Theorem 4). For $p>2$ , the model satisfies the Uniform-LGI but not PL condition;**
    + **replace the original Figure 1 with a new figure that provides numerical results for Gaussian initialization with varied mean and variance.**

  + According to Reviewer xkEt's comments, we have
     + **modified the statement of Theorem 2, and add explanations for the roles of Theorem 1 \& 2 before and after Theorem 2;**
     + **added additional experiments for different learning rates (Appendix A.1);**
     + **provided numerical experiments to show the relation between optimization paths and generalization on MNIST dataset (Appendix A.2).**

  + According to Reviewer shcV's comments, we have
     + **added explanations of the relation between Theorem 1 \& 2 before and after Theorem 2,
          for better understanding how explicitly the length of optimization path influence the generalization bound.**

---

### Decision · Program_Chairs · 2022-01-20

**Decision:**

Reject

**Comment:**

The paper presents several interesting generalization results for Uniform-LGI loss functions (a generalization of PL functions). Some of these bounds seem useful, but the overall connection with the optimization length remains unclear. This concern and other points of criticism remain present after the rebuttal phase. Other minor concerns seem fixable, but in a larger timeline compared to the camera ready one. The paper should be revised for a future venue.